# Learning Neural Set Functions Under the Optimal Subset Oracle

**Zijing Ou**[1,2], **Tingyang Xu**[1], **Qinliang Su**[3], **Yingzhen Li**[2], **Peilin Zhao**[1], **Yatao Bian**[1*]

[1]Tencent AI Lab, China
[2]Imperial College London, United Kingdom
[3]Sun Yat-sen University, China

z.ou22@imperial.ac.uk {tingyangxu,masonzhao}@tencent.com
suqliang@mail.sysu.edu.cn yingzhen.li@imperial.ac.uk yatao.bian@gmail.com

## Abstract

Learning neural set functions becomes increasingly important in many applications like product recommendation and compound selection in AI-aided drug discovery. The majority of existing works study methodologies of set function learning under the function value oracle, which, however, requires expensive supervision signals. This renders it impractical for applications with only weak supervisions under the Optimal Subset (OS) oracle, the study of which is surprisingly overlooked. In this work, we present a principled yet practical maximum likelihood learning framework, termed as EquiVSet,[1] that simultaneously meets the following desiderata of learning neural set functions under the OS oracle: i) permutation invariance of the set mass function being modeled; ii) permission of varying ground set; iii) minimum prior; and iv) scalability. The main components of our framework involve: an energy-based treatment of the set mass function, DeepSet-style architectures to handle permutation invariance, mean-field variational inference, and its amortized variants. Thanks to the elegant combination of these advanced architectures, empirical studies on three real-world applications (including Amazon product recommendation, set anomaly detection and compound selection for virtual screening) demonstrate that EquiVSet outperforms the baselines by a large margin.

## 1 Introduction

Many real-world applications involve prediction of set-value outputs, such as recommender systems which output a set of products to customers, anomaly detection that predicts the outliers from the majority of data (Zhang et al., 2020), and compound selection for virtual screening in drug discovery aims at extracting the most effective compounds from a given compound database (Gimeno et al., 2019). All of these applications implicitly learn a set function (Rezatofighi et al., 2017; Zaheer et al., 2017) that measures the utility of a given set input, such that the most desirable set output has the highest (or lowest *w.l.o.g*) utility value.

More formally, consider a recommender system: given a set of product candidates $V$, it is expected to recommend a subset of products $S^* \subseteq V$ to the user, which would satisfy the user most, *i.e.*, offering the maximum utility to the user. We assume the underlying process of determining $S^*$ can be modelled by a utility function $F_\theta(S; V)$ parameterized by $\theta$, and the following criteria:

$$S^* = \operatorname*{argmax}_{S \in 2^V} F_\theta(S; V). \tag{1}$$

---

[*]Correspondence to: Yatao Bian.
[1]Code is available at: https://github.com/SubsetSelection/EquiVSet.

There are mainly two settings for learning the utility function. The first one, namely function value (FV) oracle, targets at learning $F_\theta(S; V)$ to fit the utility explicitly, under the supervision of data in the form of $\{(S_i, f_i)\}$ for a fixed ground set $V$, where $f_i$ is the true utility function value of the subset $S_i$. However, training in this way is prohibitively expensive, since one needs to construct large amounts of supervision signals for a specific ground set $V$ (Balcan & Harvey, 2018). Here we consider an alternative setting, which learns $F_\theta(S; V)$ in an implicit way. More formally, with the data in form of $\{(V_i, S_i^*)\}_{i=1}^N$, where $S_i^*$ is the optimal subset (OS) corresponding to $V_i$, our goal is to estimate $\theta$ such that for all possible $(V_i, S_i^*)$, it satisfies equation (1). The OS oracle is arguably more practical than the FV oracle, which alleviates the need for explicitly labeling utility values for a large amount of subsets.[2]

Though being critical for practical success, related study on set utility function learning under the OS supervision oracle is surprisingly lacked. The most relevant work is the probabilistic greedy model (PGM), which solves the optimization problem of (1) with a greedy maximization algorithm (Tschiatschek et al., 2018). Specifically, PGM interprets the maximization algorithm as to construct differentiable distributions over sequences of items in an auto-regressive manner. However, such construction of distributions is problematic for defining distributions on sets due to the dependency on the sampling order. Therefore, they alleviate this issue by enumerating all possible permutations of the sampling sequence (detailed discussion is given in Appendix A). Such enumerations scale poorly due to the combinatorial cost $\mathcal{O}(|V|!)$, which hinders PGM's applicability to real-world applications.

To learn set functions under the OS oracle, we advocate the maximum likelihood paradigm (Stigler, 1986). Specifically, this learning problem can be viewed from a probabilistic perspective

$$\underset{\theta}{\mathrm{argmax}} \ \mathbb{E}_{\mathbb{P}(V,S)}[\log p_\theta(S|V)] \tag{2}$$
$$\mathrm{s.\,t.} \ p_\theta(S|V) \propto F_\theta(S; V), \forall S \in 2^V,$$

where the constraint admits the learned set function to obey the objective defined in (1). Given limited data $\{(V_i, S_i^*)\}_{i=1}^N$ sampled from the underlying data distribution $\mathbb{P}(V, S)$, one would maximize the empirical log likelihood: $\sum_{i=1}^N [\log p_\theta(S_i^*|V_i)]$. The most important step is to construct a proper set distribution $p_\theta(S|V)$ whose probability mass monotonically grows with the utility function $F_\theta(S; V)$ and satisfy the following additive requirements: (i) *permutation invariance*: the probability mass should not change under any permutation of the elements in $S$; (ii) *varying ground set*: the function should be able to process input sets of variable size; iii) *minimum prior*: we should make no assumptions of the set probability, *i.e.,* with maximum entropy, which is equivalent to the uninformative prior (Jeffreys, 1946); and iv) *scalibility*: the learning algorithm should be scalable to large-scale datasets and run in polynomial time.

In this paper, we propose **Equi**variant **V**ariational inference for **Set** function learning (EquiVSet), a new method for learning set functions under the OS oracle, which satisfies all the requirements. Specifically, we use an energy-based model (EBM) to construct the set mass function. EBMs are maximum entropy distributions, which satisfies the *minimum prior* requirement. Moreover, by modeling the energy function with DeepSet-style architectures (Zaheer et al., 2017; Lee et al., 2019), the two requirements, *i.e., permutation invariance* and *varying ground set* are naturally satisfied. Unfortunately, the flexibility of EBMs exacerbates the difficulties of learning and inference, since the inputs of set are discrete and lie in an exponentially-large space. To remedy this issue, we develop an approximate maximum likelihood approach which estimates the marginals via the mean-field variational inference, resulting in an efficient training manner under the supervision of OS oracles. In order to ensure *scalability*, an amortized inference network with permutation equivariance is proposed, which allows the model to be trained on large-scale datasets.

Although it may be seen as combining existing components in approximate inference, the proposed framework addresses a surprisingly overlooked problem in the set function learning communities using an intuitive yet effective method. Our main contributions are summarized below:

---

[2]Notably, learning set functions under the OS oracle is distinct to that under the FV oracle; the two settings are not comparable in general. To illustrate this, one can easily obtain the FV oracle of maximum cut set functions, but fail to specify the OS oracle since it is NP-complete to solve the maximum cut problem (Garey & Johnson, 1979, Appendix A2.2). Moreover, even though the OS oracle naturally shows up in the product recommendation scenario, one cannot identify its FV oracle since the true utility values are hard to obtain.

- We formulate set functions learning problems under the OS supervision oracle using the maximum likelihood principle;

- We present an elegant framework based on EBMs which satisfies the four desirable requirements and is efficient both at training and inference stages;

- Real-world experiments demonstrate effectiveness of the proposed OS learning framework.

## 2 Energy-Based Modeling for Set Function Learning

The first step to solve problem (2) is to construct a proper set mass function $p_\theta(S|V)$ monotonically growing with the utility function $F_\theta(S; V)$. There exits countless ways to construct such a probability mass function, such as the sequential modeling in PGM (Tschiatschek et al., 2018, Section 4). Here we resort to the energy-based treatment:

$$p_\theta(S|V) = \frac{\exp(F_\theta(S; V))}{Z}, \ Z := \sum_{S' \subseteq V} \exp(F_\theta(S'; V)), \tag{3}$$

where the utility function $F_\theta(S; V)$ stands for the negative energy, with higher utility representing lower energy. The energy-based treatment is attractive, partially due to its maximum entropy (*i.e.*, minimum prior) property. That is, it assumes nothing about what is unknown, which is known as the "noninformative prior" principle in Bayesian modeling (Jeffreys, 1946). This basic principle is, however, violated by the set mass function defined in PGM. We refer detailed motivation of the energy-based modeling to Appendix B.1.

In addition to the *minimum prior*, the energy-based treatment also enables the set mass function $p_\theta(S|V)$ to meet the other two requirements, *i.e. permutation invariance* and *varying ground set*, by deliberately designing a suitable set function $F_\theta(S; V)$. However, modeling such a proper function is nontrivial, since classical feed-forward neural networks (e.g., the ones designed for submodular set functions (Bilmes & Bai, 2017)) violate both two criteria, which restricts their applicability to the problems involving a set of objects. Fortunately, Zaheer et al. (2017) sidestep this issue by introducing a novel architecture, namely DeepSet. They theoretically prove the following Proposition.

**Proposition 1.** *All permutation invariant set functions can be decomposed in the form $f(S) = \rho\left(\sum_{s \in S} \kappa(s)\right)$, for suitable transformations $\kappa$ and $\rho$.*

By combining the energy-based model in (3) with DeepSet-style architectures, we could construct a valid set mass function to meet two important criteria: *permutation invaraince* and *varying ground set*. However, the flexibility of EBMs exacerbates the difficulties of learning and inference, since the partition function $Z$ is typically intractable and the input of sets is undesirably discrete.

## 3 Approximate Maximum Likelihood Learning with OS Supervision Oracle

In this section, we explore an effective framework for learning set functions under the supervision of optimal subset oracles. We start with discussing the principles for learning parameter $\theta$, followed by discussing the detailed inference method for discrete EBMs.

### 3.1 Training Discrete EBMs Under the Guidance of Variational Approximation

For discrete data, *e.g.,* set, learning the parameter $\theta$ in (3) via maximum likelihood is notoriously difficult. Although one could apply techniques, such as ratio matching (Lyu, 2012), noise contrastive estimation (Tschiatschek et al., 2016), and contrastive divergence (Carreira-Perpinan & Hinton, 2005), they generally suffer from instability on high dimensional data, especially when facing very large ground set in real-world applications. Instead of directly maximizing the log likelihood, we consider an alternative optimization objective that is computationally preferable. Specifically, we first fit a variational approximation to the EBM by solving

$$\boldsymbol{\psi}^* = \underset{\boldsymbol{\psi}}{\arg\min} \, D(q(S; \boldsymbol{\psi}) || p_\theta(S)), \tag{4}$$

| **Algorithm 1** MFVI$(\boldsymbol{\psi}, V, K)$ | **Algorithm 2** DiffMF$(V, S^*)$ | **Algorithm 3** EquiVSet$(V, S^*)$ |
|---|---|---|
| 1: **for** $k \leftarrow 1, \ldots, K$ **do** | 1: initialize variational parameter | 1: update parameter $\phi$ using (6) |
| 2:    **for** $i \leftarrow 1, \ldots, \|V\|$ in parallel **do** |    $\boldsymbol{\psi}^{(0)} \leftarrow 0.5 * \mathbf{1}$ |    $\phi \leftarrow \phi + \eta \nabla_\phi \text{ELBO}(\phi)$ |
| 3:      sample $m$ subsets | 2: compute the marginals | 2: initialize variational parameter |
|     $S_n \sim q(S; (\boldsymbol{\psi}^{(k-1)}|\psi_i^{(k-1)} \leftarrow 0))$ |    $\boldsymbol{\psi}^* \leftarrow \text{MFVI}(\boldsymbol{\psi}^{(0)}, V, K)$ |    $\boldsymbol{\psi}^{(0)} \leftarrow \text{EquiNet}(V; \phi)$ |
| 4:      update variational parameter | 3: update parameter $\theta$ using (5) | 3: one step fixed point iteration |
|   $\boldsymbol{\psi}_i^{(k)} \leftarrow \sigma(\frac{1}{m}\sum_{n=1}^{m}[F_\theta(S_n+i) - F_\theta(S_n)])$ |    $\theta \leftarrow \theta - \eta \nabla_\theta \mathcal{L}(\theta; \boldsymbol{\psi}^*)$ |    $\boldsymbol{\psi}^* \leftarrow \text{MFVI}(\boldsymbol{\psi}^{(0)}, V, K=1)$ |
| 5:    **end for** | | 4: update parameter $\theta$ using (5) |
| 6: **end for** | |    $\theta \leftarrow \theta - \eta \nabla_\theta \mathcal{L}(\theta; \boldsymbol{\psi}^*)$ |

Figure 1: The main components and algorithms in our framework. Note that `DiffMF` and `EquiVSet` are for one training sample only. Detailed and self-contained descriptions of each component of these algorithms are presented in Appendix D.

where $D(\cdot||\cdot)$ is a discrepancy measure between two distributions, $p_\theta(S)$[3] is the EBM defined in (3), and $q(S; \boldsymbol{\psi})$ denotes the mean-field variational distribution with the parameter $\boldsymbol{\psi} \in [0,1]^{|V|}$ standing for the odds that each item $s \in V$ shall be selected in the optimal subset $S^*$. Note that the optimal parameter $\boldsymbol{\psi}^*$ of (4) can be viewed as a function of $\theta$. In this regard, we can optimize the parameter $\theta$ by minimizing the following cross entropy loss,[4] which is well-known to be implementing the maximum likelihood estimation (Goodfellow et al., 2016) *w.r.t.* the surrogate distribution $q(S; \boldsymbol{\psi}^*)$,

$$\mathcal{L}(\theta; \boldsymbol{\psi}^*) = \mathbb{E}_{\mathbb{P}(V,S)}[-\log q(S; \boldsymbol{\psi}^*)] \approx \frac{1}{N}\sum_{i=1}^{N}\left(-\sum_{j \in S_i^*}\log \psi_j^* - \sum_{j \in V_i \setminus S_i^*}\log(1 - \psi_j^*)\right). \quad (5)$$

This is also known as the marginal-based loss (Domke, 2013), which trains probabilistic models by evaluating them using the marginals approximated by an inference algorithm. Despite not exactly bounding the log-likelihood of (3), this objective, as pointed out by Domke (2013), benefits from taking the approximation errors of inference algorithm into account while learning. However, minimizing (5) requires the variational parameter $\boldsymbol{\psi}^*$ being differentiable *w.r.t.* $\theta$. Inspired by the differentiable variational approximation to the Markov Random Fields (Krähenbühl & Koltun, 2013; Zheng et al., 2015; Dai et al., 2016), below, we extend this method to the deep energy-based formulation, which admits an end-to-end training paradigm with the back-propagation algorithm.

### 3.2 Differentiable Mean Field Variational Inference

To solve the optimization problem (4), we need to specify the variational distribution $q(S; \boldsymbol{\psi})$ and the divergence measure $D(\cdot||\cdot)$, such that the optimum marginal $\boldsymbol{\psi}^*$ is differentiable *w.r.t.* the model parameter $\theta$. A natural choice is to restrain $q(S; \boldsymbol{\psi})$ to be fully factorizable, which leads to a mean-field approximation of $p_\theta(S)$. The simplest form of $q(S; \boldsymbol{\psi})$ would be a $|V|$ independent Bernoulli distribution, *i.e.*, $q(S; \boldsymbol{\psi}) = \prod_{i \in S} \psi_i \prod_{i \notin S}(1 - \psi_i), \boldsymbol{\psi} \in [0,1]^{|V|}$. Further restricting the discrepancy measure $D(q||p)$ to be the Kullback-Leibler divergence, we recover the well-known mean-field variational inference method. It turns out that minimizing the KL divergence amounts to maximizing the evidence lower bound (ELBO)

$$\min_{\boldsymbol{\psi}} \mathbb{KL}(q(S, \boldsymbol{\psi})||p_\theta(S)) \quad \Leftrightarrow \quad \max_{\boldsymbol{\psi}} f_{\text{mt}}^{F_\theta}(\boldsymbol{\psi}) + \mathbb{H}(q(S; \boldsymbol{\psi})) =: \text{ELBO}, \quad (6)$$

where $f_{\text{mt}}^{F_\theta}(\boldsymbol{\psi})$ is the multilinear extension of $F_\theta(S)$ (Calinescu et al., 2007), which is defined as

$$f_{\text{mt}}^{F_\theta}(\boldsymbol{\psi}) := \sum_{S \subseteq V} F_\theta(S) \prod_{i \in S} \psi_i \prod_{i \notin S}(1 - \psi_i), \boldsymbol{\psi} \in [0,1]^{|V|}. \quad (7)$$

To maximize the ELBO in (6), one can apply the fixed point iteration algorithm. Specifically, for coordinate $\psi_i$, the partial derivative of the multilinear extension is $\nabla_{\psi_i} f_{\text{mt}}^{F_\theta}(\boldsymbol{\psi})$, and for the

---

[3]Here we omit the condition $V$ for brevity. In some specific context, it would be helpful to regard subset $S$ as a binary vector, *i.e.*, $S := \{0, 1\}^{|V|}$ with the $i$-th element equal to 1 meaning $i \in S$ and 0 meaning $i \notin S$.

[4]This objective would suffer from label-imbalanced problem when the size of OS is too small. In practice, we can apply negative sampling to overcome this problem: we randomly select a negative set $N_i \subseteq V_i \setminus S_i^*$ with the size of $|S^*|$, and train the model with an alternative objective $\sum_i -\sum_{j \in S_i^*}\log \psi_j^* - \sum_{j \in N_i}\log(1 - \psi_j^*)$.

entropy term, it is $\nabla_{\psi_i}\mathbb{H}(q) = \log\frac{1-\psi_i}{\psi_i}$. Thus, the stationary condition of maximizing ELBO is $\psi_i = \sigma(\nabla_{\psi_i}f_{\mathrm{mt}}^{F_\theta}(\boldsymbol{\psi})), i = 1, \ldots, |V|$, where $\sigma$ is the sigmoid function, which means $\psi_i$ should be updated as $\psi_i \leftarrow \sigma(\nabla_{\psi_i}f_{\mathrm{mt}}^{F_\theta}(\boldsymbol{\psi}))$. This analysis leads to the traditional mean field iteration, which updates each coordinate one by one (detailed derivation in Appendix B.2). In this paper, we suggest to update $\boldsymbol{\psi}$ in a batch manner, which is more efficient in practice. More specifically, we summerize the mean field approximation as the following fixed-point iterative update steps

$$\boldsymbol{\psi}^{(0)} \leftarrow \text{Initialize in } [0,1]^{|V|}, \tag{8}$$

$$\boldsymbol{\psi}^{(k)} \leftarrow (1 + \exp(-\nabla_{\boldsymbol{\psi}^{(k-1)}}f_{\mathrm{mt}}^{F_\theta}(\boldsymbol{\psi}^{(k-1)})))^{-1}, \tag{9}$$

$$\boldsymbol{\psi}^* \leftarrow \boldsymbol{\psi}^{(K)}. \tag{10}$$

We denote the above iterative steps as a function termed as $\mathrm{MFVI}(\boldsymbol{\psi}, V, K)$, which takes initial vairational parameter $\boldsymbol{\psi}$, ground set $V$, and number of iteration steps $K$ as input, and outputs the parameter $\boldsymbol{\psi}^*$ after $K$ steps. Note that, $\mathrm{MFVI}(\boldsymbol{\psi}, V, K)$ is differentiable *w.r.t.* the parameter $\theta$, since each fixed-point iterative update step is differentiable. Thereby, one could learn $\theta$ by minimizing the cross entropy loss in (5). However, the computation complexity raises from the derivative of multilinear extension $f_{\mathrm{mt}}^{F_\theta}(\boldsymbol{\psi})$ defined in (7), which sums up all the possible subsets in the space of size $2^{|V|}$. Fortunately, the gradient $\nabla_{\boldsymbol{\psi}}f_{\mathrm{mt}}^{F_\theta}$ can be estimated efficiently via Monte Carlo approximation methods, since the following equation holds.

$$\nabla_{\psi_i}f_{\mathrm{mt}}^{F_\theta}(\boldsymbol{\psi}) = \mathbb{E}_{q(S;(\boldsymbol{\psi}|\psi_i\leftarrow 0))}\left[F_\theta(S+i) - F_\theta(S)\right], \tag{11}$$

in which we use $S+i$ to denote the set union $S \cup \{i\}$. Detailed derivation is provided in Appendix B.3. According to (11), we can estimate the partial derivative $\nabla_{\psi_i}f_{\mathrm{mt}}^{F_\theta}$ via Monte Carlo approximation: i) sample $m$ subsets $S_n, n = 1, \ldots, m$ from the surrogate distribution $q(S; (\boldsymbol{\psi}|\psi_i \leftarrow 0))$; ii) approximate the expectation by the average $\frac{1}{m}\sum_{k=1}^{n}[F_\theta(S_n+i) - F_\theta(S_n)]$. After training, the OS for a given ground set can be sampled via rounding $\boldsymbol{\psi}^*$, which is the optimal variational parameter after $K$-steps mean-field iteration, *i.e.,* $\boldsymbol{\psi}^* = \mathrm{MFVI}(\boldsymbol{\psi}, V, K)$, and stands for the probability of each element in the ground set should be sampled.[5] We term this method as **Diff**erentiable **M**ean **F**ield (DiffMF) and summarize the training and inference process in Algorithm 2 and 1, respectively.

## 4 Amortizing Inference with Equivariant Neural Networks

Although DiffMF can learn set function $F_\theta$ in an effective way, it undesirably has two notorious issues: i) the computation is in general prohibitively expensive, since DiffMF involves a typically expensive sampling loop per data point; ii) some information regarding interactions between elements is discarded, since DiffMF assumes a fully fatorizable variational distribution. In this section, we first propose to amortize the inference process with an additional recognition neural network, and then extend it to considering correlation for more accurate approximations.

### 4.1 Equivariant Amortized Variational Inference

To enable training the proposed model on a large-scale dataset, we propose to amortize the approximate inference process with an additional recognition neural network which outputs parameter $\boldsymbol{\psi}$ for the variational distribution $q_\phi(S; \boldsymbol{\psi})$,[6] where $\phi$ denotes the parameter of neural networks. A proper recognition network involving set objects shall satisfy the property of *permutation equivariance*.

**Definition 1.** *A function $f : \mathcal{X}^d \to \mathcal{Y}^d$ is called permutation equivalent when upon permutation of the input instances permutes the output labels, i.e., for any permutation $\pi$: $f(\pi([x_1, \ldots, x_d])) = \pi(f([x_1, \ldots, x_d]))$.*

Zaheer et al. (2017) propose to formulate the *permutation equivariant* architecture as :

$$f_i(S) = \rho\left(\lambda\kappa(s_i) + \gamma\sum_{s\in S}\kappa(s)\right), \tag{12}$$

---

[5]Here we simply apply the topN rounding, but it is worthwhile to explore other rounding methods as a future work.

[6]With a slight abuse of notations, we use the same symbol here as in (6).

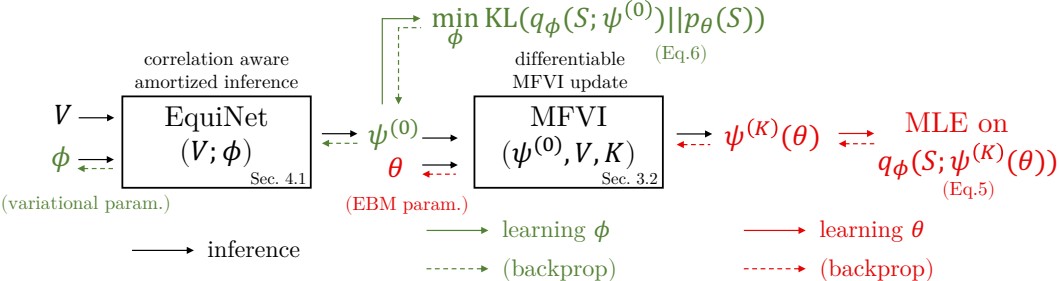

Figure 2: Overview of the training and inference processes of `EquiVSet`.

where $s_i$ denotes the $i^{\text{th}}$ element in the set $S$, $\lambda, \gamma$ are learnable scalar variables, and $\rho, \kappa$ are any proper transformations. Note that the output value of $f_i : 2^V \to [0,1]$ is relative to the $i^{\text{th}}$ coordinate, but not the order of the elements in $S$. Thus the equivariant recognition network, denoted as $\psi = \text{EquiNet}(V; \phi) : 2^V \to [0,1]^{|V|}$, can be defined as $\text{EquiNet}_i := f_i$, which takes the ground set $V$ as input and outputs the distribution parameter $\psi$ for $q_\phi(S; \psi)$.

### 4.2 Correlation-aware Inference with Gaussian Copula

Due to the mean-field assumption, the proposed variational distribution cannot model the interactions among elements in the input set. We address this issue by introducing Gaussian copula (Nelsen, 2007), which is a cumulative distribution function (CDF) of random variables $(u_1, \ldots, u_{|V|})$ over the unit cube $[0,1]^{|V|}$, with $u_i \sim \text{Uniform}(0,1)$. More formally, given a covariance matrix $\mathbf{\Sigma}$, the Gaussian copula $C_{\mathbf{\Sigma}}$ with parameter $\mathbf{\Sigma}$ is defined as

$$C_{\mathbf{\Sigma}}(u_1, \cdots, u_{|V|}) = \Phi_{\mathbf{\Sigma}}\left(\Phi^{-1}(u_1), \cdots, \Phi^{-1}(u_{|V|})\right),$$

where $\Phi_{\mathbf{\Sigma}}$ stands for the joint CDF of a Gaussian distribution with zero mean and covariance matrix $\mathbf{\Sigma}$, and $\Phi^{-1}$ is the inverse CDF of standard Gaussian. With the location parameter $\psi$ output by $\text{EquiNet}(V; \phi)$, we can induce correlation into the Bernoulli distribution via the following way: i) sample an auxiliary noise $\boldsymbol{g} \sim \mathcal{N}(\mathbf{0}, \mathbf{\Sigma})$; ii) apply element-wise Gaussian CDF $\boldsymbol{u} = \mathbf{\Phi}_{\text{diag}(\mathbf{\Sigma})}(\boldsymbol{g})$; iii) obtain binary sample via $\boldsymbol{s} = \mathbb{I}(\psi \leq \boldsymbol{u})$,[7] where $\psi \leq \boldsymbol{u}$ means $\forall i, \psi_i \leq u_i$, $\mathbb{I}(\cdot)$ is the indicator function, and $\text{diag}(\mathbf{\Sigma})$ returns the diagonal matrix of $\mathbf{\Sigma}$. In practice, the covariance matrix $\mathbf{\Sigma}$ could be generated by another neural network with the input ground set. We refer the discussion on it to Appendix C, and demonstrate how to efficiently construct and sample from a non-diagonal Gaussian distribution, while retaining a *permutation equivariant* sampling process.

To learn the parameters of the variational distribution, one can maximize the ELBO objective in (6). However, the ELBO has no differentiable closed-form expression *w.r.t.* $\phi$.[8] To remedy this, we relax the binary variable $\boldsymbol{s}$ to a continuous one by applying the Gumbel-Softmax trick (Jang et al., 2016), resulting in an end-to-end training process with backpropagation.

### 4.3 Details of Training and Inference

Our model consists of two components: the EBM $p_\theta(S)$ and the variational distribution $q_\phi(S; \psi)$. As shown in Figure 2, these two components are trained in a cooperative learning fashion (Xie et al., 2018). Specifically, we train the variational distribution $q_\phi$ with fixed $\theta$ firstly by maximizing the ELBO in (6). To train the energy model $p_\theta$, we first initialize the variational parameter $\psi^{(0)}$ with the output of equivariant recognition network $\text{EquiNet}(V; \phi)$. This enables us to get a more accurate variational approximate, since $q_\phi$ has modeled the correlation among the elements in the set. Notice that $\psi^{(0)}$ does not depend on $\theta$ directly. To learn $\theta$, we take one further step of mean-field iteration $\text{MFVI}(\psi^{(0)}, V, 1)$, which flows the gradient through $\theta$ and enables to optimize $\theta$ using the cross entropy loss in (5) (*i.e.,* if we skip step 3 in Algorithm 3, and feed $\psi^{(0)}$ to step 4, the gradient would not flow through $\theta$). However, if we take multiple steps, it inclines to converge to the local optima that

---

[7]Here $\boldsymbol{s}$ is a binary vector $\{0,1\}^{|V|}$ with the $i$-th element equal to 1 meaning $i \in S$ and 0 meaning $i \notin S$.

[8]For correlation-aware inference, the variational parameter $\phi$ consists of two parts: i) $\phi$ of the $\text{EquiNet}(V; \phi)$ and ii) $\mathbf{\Sigma}$ of the Gaussian copula.

is the same as the original mean-field iteration. As a result, the benefit of correlation-aware inference provided by the Gaussian copula would be diminished. Detailed analysis is provided in Appendix F.5. The training procedure is summarized in Algorithm 3 (the complete version is given in Appendix D).

For *inference* in the test time, given a ground set $V$, we initialize the variational parameter via $\boldsymbol{\psi}^{(0)} = \text{EquiNet}(V; \phi)$, then run one step mean-field iteration $\boldsymbol{\psi}^* \leftarrow \text{MFVI}(\boldsymbol{\psi}^{(0)}, V, 1)$. Finally, the corresponding OS is obtained by applying the topN rounding method. We term our method as **Equi**variant **V**ariational Inference for **Set** Function Learning (`EquiVSet`), and respectively use `EquiVSet`$_{\text{ind}}$ and `EquiVSet`$_{\text{copula}}$ to represent two variants with independent and copula variational posterior, respectively.

## 5 Related Work

**Set function learning.** There is a growing literature on learning set functions with deep neural networks. Zaheer et al. (2017) designed the DeepSet architecture to create permutation invariant and equivariant function for set prediction. Lee et al. (2019) enhanced model ability of DeepSet by employing transformer layer to introduce correlation among instances of set, and Horn et al. (2020) extended this framework for time series. It is noteworthy that they all learn set functions under the function value oralce and can be employed as the backbone of the utility function $F_\theta(S; V)$ in our model. Dolhansky & Bilmes (2016); Bilmes & Bai (2017); Ghadimi & Beigy (2020) have also designed deep architectures for submodular set functions, however, these designs can not handle the varying ground set requirement. There are papers studying the learnability of specific set functions (e.g., submodular functions and subadditive functions) in a distributional learning setting (Balcan et al., 2012; Badanidiyuru et al., 2012; Balcan & Harvey, 2018) under the function value oracle, they mainly provide sample complexity with inapproximability results under the probably mostly approximately correct (PMAC) learning model. Other methods relevant to our setting are TSPN (Kosiorek et al., 2020) and DESP (Zhang et al., 2020). However they both focused on generating set objects under a given condition. While we aim at predicting under the optimal subset oracle.

**Energy-based modeling.** Energy based learning (LeCun et al., 2006) is a classical framework to model the underlying distribution over data. Since it makes no assumption of data, energy-based models are extremely flexible and have been applied to wide ranges of domains, such as data generation (Nijkamp et al., 2019), out-of-distribution detection (Liu et al., 2020), game-theoretic valuation algorithms (Bian et al., 2022) and biological structure prediction (Shi et al., 2021). Learning EBMs can be done by applying some principled methods, like contrastive divergence (Hinton, 2002), score matching (Hyvärinen & Dayan, 2005), and ratio matching (Lyu, 2012). For inference, gradient-based MCMC methods (Welling & Teh, 2011; Grathwohl et al., 2021) are widely exploited. Meanwhile, Bian et al. (2019); Sahin et al. (2020) propose provable mean-filed inference algorithms for a class of EBMs with supermodular energies (also called probabilistic log-submodular models). In this paper, we train EBMs under the supervision of OS oracle by running mean-field inference.

**Amortized and Copula variational inference.** Instead of approximating separate variables for each data point, amortized variational inference (VI) (Kingma & Welling, 2013) assumes that the variational parameters can be predicted by a parameterized function of the data (Zhang et al., 2018). The idea of amortized VI has been widely applied in deep probabilistic models (Hoffman et al., 2013; Garnelo et al., 2018). Although this procedure would introduce an amortization gap (Cremer et al., 2018), which refers to the suboptimality of variational parameters, amortized VI enables significant speedups and combines probabilistic modeling with the representational power of deep learning. Copula is the other method to improve the representational power for VI. Tran et al. (2015) used copula to augment the mean-field VI for better posterior approximation. Suh & Choi (2016) adopted Gaussian copula in VI to model the dependency structure of observed data. However, none of them can be directly applied to our setting involving discrete latent variables.

## 6 Empirical Studies

We evaluate the proposed methods on various tasks: product recommendation, set anomaly detection, compound selection, and synthetic experiments. All experiments are repeated five times with different random seeds and their means and standard deviations are reported. The model architectures and training details are deferred to Appendix E. Additional experiments of varying ground set are given

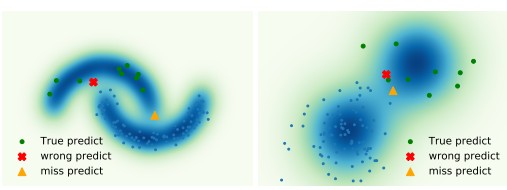

Figure 3: Visualization of the prediction of $\texttt{EquiVSet}_{\text{copula}}$ on the Two-Moons (left) and Gaussian-Mixture (right) datasets.

Table 1: Results in the MJC metric on Two-Moons and Gaussian-Mixture datasets.

| Method | Two Moons | Gaussian Mixture |
|---|---|---|
| Random | 0.055 | 0.055 |
| PGM | $0.360 \pm 0.020$ | $0.438 \pm 0.009$ |
| DeepSet (NoSetFn) | $0.472 \pm 0.003$ | $0.446 \pm 0.002$ |
| $\texttt{DiffMF}$ (ours) | $0.584 \pm 0.001$ | $0.908 \pm 0.002$ |
| $\texttt{EquiVSet}_{\text{ind}}$ (ours) | $0.570 \pm 0.003$ | $0.907 \pm 0.002$ |
| $\texttt{EquiVSet}_{\text{copula}}$ (ours) | $\mathbf{0.587 \pm 0.002}$ | $\mathbf{0.909 \pm 0.002}$ |

in Appendix F.1. Comparisons with Set Transformer (Lee et al., 2019) are in Appendix F.2. Ablation studies on hyper-parameter choices (e.g. MFVI iteration steps, number of MC samples, rank of perturbation, temperature of Gumbel-Softmax) are provided in Appendix F.5.

**Evaluations.** We evaluate the methods using the mean Jaccard coefficient (MJC). Specifically, for each sample $(V, S^*)$, denoting the corresponding model predict as $S'$, the Jaccard coefficient is defined as $\text{JC}(S, S') = \frac{|S' \cap S|}{|S' \cup S|}$. Then the MJC metric can be computed by averaging over all samples in the test set: $\text{MJC} = \frac{1}{|\mathcal{D}_t|} \sum_{(V, S^*) \in \mathcal{D}_t} \text{JC}(S^*, S')$.

**Baselines.** We compare our solution variants, *i.e.*, $\texttt{DiffMF}$, $\texttt{EquiVSet}_{\text{ind}}$, and $\texttt{EquiVSet}_{\text{copula}}$ to the following three baselines:

- Random: The expected performance of random guess. This baseline provides an estimate of how difficult the task is. Specifically, given a data point $(V, S^*)$, it can be computed as $\mathbb{E}(JC(V, S^*)) = \sum_{k=0}^{|S^*|} \frac{\binom{|S^*|}{k}\binom{|V|-|S^*|}{|S^*|-k}}{\binom{|V|}{|S^*|}} \frac{k}{2|S^*|-k}$.

- PGM (Tschiatschek et al. (2018), see Appendix A): The probabilistic greedy model, which is permutation invariant but computationally prohibitive.

- DeepSet (NoSetFn) (Zaheer et al., 2017): The deepset architecture, satisfying permutation invariant, is the backbone of our models. Its adapted version: $2^V \to [0, 1]^{|V|}$, which serves as the amortized networks in $\texttt{EquiVSet}$, could work as a baseline since its output stands for the probability of which instance should be selected. We train it with cross entropy loss and sample the subset via topN rounding. The term "NoSetFn" is used to emphasize that this baseline does not learn a set function explicitly, although it can be adapted to our empirical studies.

**Synthetic Experiments.** We demonstrate the effectiveness of our models on learning set functions with two synthetic datasets: the two-moons dataset with additional noise of variance $\sigma^2 = 0.1$, and mixture of Gaussians $\frac{1}{2}\mathcal{N}(\boldsymbol{\mu}_0, \boldsymbol{\Sigma}) + \frac{1}{2}\mathcal{N}(\boldsymbol{\mu}_1, \boldsymbol{\Sigma})$, with $\boldsymbol{\mu}_0 = [\frac{1}{\sqrt{2}}, \frac{1}{\sqrt{2}}]^T$, $\boldsymbol{\mu}_1 = -\boldsymbol{\mu}_0$, $\boldsymbol{\Sigma} = \frac{1}{4}\mathbf{I}$. Take the Gaussian mixture as an example, the data generation procedure is as follow: i) select index: $b \sim Bernoulli(\frac{1}{2})$; ii) sample 10 points from $\mathcal{N}(\boldsymbol{\mu}_b, \boldsymbol{\Sigma})$ to construct $S^*$; iii) sample 90 points for $V \backslash S^*$ from $\mathcal{N}(\boldsymbol{\mu}_{1-b}, \boldsymbol{\Sigma})$. We collect $1,000$ samples for training, validation, and test, respectively.

A qualitative result of the $\texttt{EquiVSet}_{\text{copula}}$ is shown in Figure 3, where the green dots represent correct model predictions, the red crosses are incorrect model predictions, and the yellow triangles represent the data points in the subset oracle $S^*$ that are missed by the model. One can see that the most confusing points are located at the intersection of two components. We also illustrate the quantitative results in Table 1. As expected, our methods achieve significantly better performance over other methods, with averaged $59.16\%$ and $100.07\%$ improvements compared to PGM on the Two-Moons and Gaussian-Mixture datasets, respectively.

**Product Recommendation.** In this experiment, we use the Amazon baby registry dataset (Gillenwater et al., 2014), which contains numerous subsets of products selected by different customers. Amazon characterizes each product in a baby registry as belonging to a specific category, such as "toys" and "furniture". Each product is characterized by a short textual description and we represent it as a 768 dimensional vector using the pre-trained BERT model (Devlin et al., 2018).

Table 2: Product recommendation results on the Amazon dataset with different categories.

| Categories | Random | PGM | DeepSet (NoSetFn) | DiffMF (ours) | EquiVSet$_{\text{ind}}$ (ours) | EquiVSet$_{\text{copula}}$ (ours) |
|---|---|---|---|---|---|---|
| Toys | 0.083 | $0.441 \pm 0.004$ | $0.429 \pm 0.005$ | $0.610 \pm 0.010$ | $0.650 \pm 0.015$ | $\mathbf{0.680 \pm 0.020}$ |
| Furniture | 0.065 | $0.175 \pm 0.007$ | $\mathbf{0.176 \pm 0.007}$ | $0.170 \pm 0.010$ | $0.170 \pm 0.011$ | $0.172 \pm 0.009$ |
| Gear | 0.077 | $0.471 \pm 0.004$ | $0.381 \pm 0.002$ | $0.560 \pm 0.020$ | $0.610 \pm 0.020$ | $\mathbf{0.700 \pm 0.020}$ |
| Carseats | 0.066 | $\mathbf{0.230 \pm 0.010}$ | $0.210 \pm 0.010$ | $0.220 \pm 0.010$ | $0.214 \pm 0.007$ | $0.210 \pm 0.010$ |
| Bath | 0.076 | $0.564 \pm 0.008$ | $0.424 \pm 0.006$ | $0.690 \pm 0.006$ | $0.650 \pm 0.020$ | $\mathbf{0.757 \pm 0.009}$ |
| Health | 0.076 | $0.449 \pm 0.002$ | $0.448 \pm 0.004$ | $0.565 \pm 0.009$ | $0.630 \pm 0.020$ | $\mathbf{0.700 \pm 0.020}$ |
| Diaper | 0.084 | $0.580 \pm 0.009$ | $0.457 \pm 0.005$ | $0.700 \pm 0.010$ | $0.730 \pm 0.020$ | $\mathbf{0.830 \pm 0.010}$ |
| Bedding | 0.079 | $0.480 \pm 0.006$ | $0.482 \pm 0.008$ | $0.641 \pm 0.009$ | $0.630 \pm 0.020$ | $\mathbf{0.770 \pm 0.010}$ |
| Safety | 0.065 | $0.250 \pm 0.006$ | $0.221 \pm 0.004$ | $0.200 \pm 0.050$ | $0.230 \pm 0.030$ | $\mathbf{0.250 \pm 0.030}$ |
| Feeding | 0.093 | $0.560 \pm 0.008$ | $0.430 \pm 0.002$ | $0.750 \pm 0.010$ | $0.696 \pm 0.006$ | $\mathbf{0.810 \pm 0.007}$ |
| Apparel | 0.090 | $0.533 \pm 0.005$ | $0.507 \pm 0.004$ | $0.670 \pm 0.020$ | $0.650 \pm 0.020$ | $\mathbf{0.750 \pm 0.010}$ |
| Media | 0.094 | $0.441 \pm 0.009$ | $0.420 \pm 0.010$ | $0.510 \pm 0.010$ | $0.551 \pm 0.007$ | $\mathbf{0.570 \pm 0.010}$ |

For each category, we generate samples $(V, S^*)$ as follows. Firstly, we filter out those subsets selected by customers whose size is equal to 1 or larger than 30. Then we split the remaining subset collection $\mathcal{S}$ into training, validation and test folds with a $1 : 1 : 1$ ratio. Finally for each OS oracle $S^* \in \mathcal{S}$, we randomly sample additional $30 - |S^*|$ products from the same category to construct $V \setminus S^*$. In this way, we construct one data point $(V, S^*)$ for each customer, which reflects this real world scenario: $V$ contains 30 products displayed to the customer, and the customer is interested in checking $|S^*|$ of them. Note that this curation process is different from that of (Tschiatschek et al., 2018, Section 5.3), which is deviated from the real world scenario (Detailed discussion in Appendix E.5.).

The performance of all the models on different categories are shown in Table 2. Evidently, our models perform favorably to the baselines. Compared with PGM, which learns the set function via a probabilistic greedy algorithm, we can observe that our models, which model the the set functions with energy-based treatments, achieves better results on all settings. Although DeepSet is also permutation invariant, our model still outperforms it by a substantial margin, indicating the superiority of learning the set function explicitly.

**Set Anomaly Detection.** In this experiment, we evaluate our methods on two image datasets: the double MNIST (Sun, 2019) and the CelebA (Liu et al., 2015b). For each dataset, we randomly split the training, validation, and test set to the size of $10,000$, $1,000$, and $1,000$, respectively.

**Double MNIST:** The dataset consists of 1000 images for each digit ranging from 00 to 99. For each sample $(V, S^*)$, we randomly sample $n \in \{2, \ldots, 5\}$ images with the same digit to construct the OS oracle $S^*$, and then select $20 - |S^*|$ images with different digits to construct the set $V \setminus S^*$. **CelebA:** The CelebA dataset contains $202,599$ images with 40 attributes. We select two attributes at random and construct the set with the size of 8. For each ground set $V$, we randomly select $n \in \{2, 3\}$ images as the OS oracle $S^*$, in which neither of the two attributes is present. See Figure 4 and Figure 5 in Appendix E.6 for illustrations of sampled data.

From Table 3, we see that the variants of our model consistently outperform baseline methods strongly. Furthermore, we observe that by introducing the correlation to the variational distribution, significant performance gains can be obtained, demonstrating the benefits of relaxing the independent assumption by using Gaussian copula. Additional experiments on the other two datasets F-MNIST (Xiao et al., 2017) and CIFAR-10 (Krizhevsky et al., 2009) are provided in Appendix F.3.

**Compound Selection in AI-aided Drug Discovery.** A critical step in drug discovery is to select compounds with high biological activity (Wallach et al., 2015; Li et al., 2021; Ji et al., 2022), diversity and satisfactory ADME (absorption, distribution, metabolism, and excretion) properties (Gimeno et al., 2019). As a result, virtual screening is typically a hierarchical filtering process with several necessary filters, e.g., first choosing the highly active compounds, then selecting diverse subsets from them, and finally excluding compounds that are bad for ADME. We finally arrive at a compound subset after a series of these steps. Given the OS supervision signals, we can learn to conduct this complicated selection process in an end to end manner. As a result, it will eliminate the need for intermediate supervision signals, which can be very expensive or impossible to obtain due to pharmacy's personal protection policy. For example, measuring the bioactivity and ADME properties of a compound has to be done in wet labs, and pharmaceutical companies are reluctant to disclose

Table 3: Set anomaly detection results.

| Method | Double MNIST | CelebA |
|---|---|---|
| Random | 0.082 | 0.219 |
| PGM | $0.300 \pm 0.010$ | $0.481 \pm 0.006$ |
| DeepSet (NoSetFn) | $0.111 \pm 0.003$ | $0.390 \pm 0.010$ |
| DiffMF (ours) | $\mathbf{0.610 \pm 0.010}$ | $0.546 \pm 0.008$ |
| EquiVSet$_{\text{ind}}$ (ours) | $0.410 \pm 0.010$ | $0.530 \pm 0.010$ |
| EquiVSet$_{\text{copula}}$ (ours) | $0.588 \pm 0.007$ | $\mathbf{0.555 \pm 0.005}$ |

Table 4: Compound selection results.

| Method | PDBBind | BindingDB |
|---|---|---|
| Random | 0.073 | 0.027 |
| PGM | $0.350 \pm 0.009$ | $0.176 \pm 0.006$ |
| DeepSet (NoSetFn) | $0.319 \pm 0.003$ | $0.162 \pm 0.007$ |
| DiffMF (ours) | $\mathbf{0.360 \pm 0.010}$ | $0.189 \pm 0.002$ |
| EquiVSet$_{\text{ind}}$ (ours) | $0.355 \pm 0.005$ | $\mathbf{0.190 \pm 0.003}$ |
| EquiVSet$_{\text{copula}}$ (ours) | $0.354 \pm 0.008$ | $0.188 \pm 0.003$ |

the data. Here we simulate the OS oracle of compound selection by applying the *two filters*: high bioactivity and diversity filters, based on the following two datasets.

**PDBBind (Liu et al., 2015a):** This dataset consists of experimentally measured binding affinities for bio-molecular complexes. We construct our dataset using the "refined" subsets therein, which contains 179 protein-ligand complexes. **BindingDB**[9]**:** It is a public database of measured binding affinities, which consists of $52,273$ drug-targets with small, drug-like molecules. Instead of providing complexes, here only the target amino acid sequence and compound SMILES string are provided.

We apply the same filtering process to construct samples $(V, S^*)$ for these two datasets. Specifically, we first randomly select a number of compounds to construct the ground set $V$, whose size is 30 and 300 for PDBBind and BindingDB, respectively. Then $\frac{1}{3}$ compounds with the highest bioactivity are filtered out, accompanied by a distance matrix measured by the corresponding fingerprint similarity of molecules. To ensure diversity, the OS oracle $S^*$ is generated by the centers of clusters which are presented by applying the affinity propagation algorithm. We finally obtain the training, validation, and test set with the size of 1,000, 100, and 100, respectively, for both two datasets. Detailed description is provided in Appendix E.7.

From Table 4, one can see that our methods magnificently outperform the random guess. This indicates that the proposed EquiVSet framework has great potential for drug discovery to facilitate the virtual screening task by modeling the complicated hierarchical selection process. Besides, improvements of EquiVSet can be further observed by comparing with DeepSet, which simply equips the deepset architecture with cross entropy loss, illustrating the superiority of explicit set function learning and energy-based modeling. Although comparable results could be achieved by PGM with sequential modeling, which satisfies permutation invariance and differentiability, our models still outperform it. This is partially because our models additionally maintain the other three desiderata of learning set functions, *i.e.,* varying ground set, minimum prior, and scalability. We also conduct a fairly simple task in Appendix F.4, in which only the bioactivity filter is considered. To simulate the full selection process, we leave it as important future work due to limited labels.

## 7    Discussion and Conclusion

We proposed a simple yet effective framework for set function learning under the OS oracle. By formulating the set probability with energy-based treatments, the resulting model enjoys the virtues of *permutation invariance*, *varying ground set*, and *minimum prior*. A *scalable* training and inference algorithm is further proposed by applying maximum log likelihood principle with the surrogate of mean-field inference. Real-world applications confirm the effectiveness of our approaches.

**Limitations & Future Works.** The training objective in (5) does not bound the log-likelihood of EBMs. A more principled discrete EBMs trainer is worth exploring. In addition, the proposed framework has the potential to facilitate learning to select subsets for other applications (Iyer et al., 2021), including active learning (Kothawade et al., 2021), targeted selection of subsets, selection of subsets for robustness (Killamsetty et al., 2020), and selection of subsets for fairness. Though we consider learning generic neural set functions in this work, it is beneficial to consider building useful priors into the neural set function architectures, such as set functions with the diminishing returns prior (Bilmes & Bai, 2017) and the bounded curvature/submodularity ratio prior (Bian et al., 2017).

---

[9]We take the curated one from https://tdcommons.ai/multi_pred_tasks/dti/

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
