# Appendix for "Learning Neural Set Functions Under the Optimal Subset Oracle"

## Contents

## A   Details of the Probabilistic Greedy Model

The probabilistic greedy model (PGM) solves optimization (1) with a differentiable extension of greedy maximization algorithm (Tschiatschek et al., 2018). Specifically, denote the first $j$ chosen elements as $S_j = \{s_1, \ldots, s_j\} \subseteq V$, PGM samples the $(j + 1)^{\text{th}}$ element from the candidate set $V \backslash S_j$ with the probability proportional to $\exp(F_\theta(s_{j+1} + S_j)/\gamma)$, which raises the probability of the selected elements in the sequence $\pi = \{s_1, s_2, \ldots, s_k\}$ as

$$p_\theta(\pi|V) = \prod_{j=0}^{k-1} \frac{\exp(F_\theta(s_{j+1} + S_j)/\gamma)}{\sum_{s \in V \backslash S_j} \exp(F_\theta(s + S_j)/\gamma)}, \tag{13}$$

where $\gamma$ is a temperature parameter, $S_0 = \emptyset$, and $s + S := S \cup \{s\}$. Note that, the computation of $p_\theta(\pi|V)$ depends on the order of sequence $\pi$, which would make the learned parameter $\theta$ sensitive to the sampling order. To alleviate this problem, Tschiatschek et al. (2018) finally construct the set mass function by enumerating all possible permutations

$$p_\theta(S|V) = \sum_{\pi \in \Pi^S} p_\theta(\pi|V), \tag{14}$$

where $\Pi^S$ is the permutation space generated from $S$. After training, the OS oracle $S$ can be sampled via sequential decoupling $p(s_{i+1}|S_i) \propto \exp(F_\theta(s_{i+1} + S_i)/\gamma)$. However, maximizing the log likelihood of (14) is prohibitively expensive and unscalable due to the exponential time complexity of enumerating all permutations. Although one can apply Monte Carlo approximation to avoid that, *i.e.,* approximating $\log p_\theta(S|V) = \log \sum_{\pi \in \Pi^S} p_\theta(\pi|V)$ with $\log p_\theta(\pi|V), \pi \sim \Pi^S$, such a simple estimator is biased, resulting in a permutation variant model.

## B  Derivations

### B.1  Derivations of the Maximum Entropy Distribution

The first step to solve problem (2) is to construct a proper set mass function $p_\theta(S|V)$ monotonically growing with the utility function $F_\theta(S; V)$. There exits countless ways to construct such a probability mass function, such as $p_\theta(S|V) \propto F_\theta(S; V)$ and the set mass function defined in PGM, *i.e.,* Equation (14). Here, one would care about what the most appropriate set mass function should be? Generally we prefer the model to assume nothing about what is unknown. More formally, we should choose the most "uniform" distribution, which maximizes the Shannon entropy $\mathbb{H}(p) = -\sum_{S \subseteq V} p(S) \log p(S)$. This principle is known as "noninformative prior" (Jeffreys, 1946), which has been widely applied in many physical systems (Jaynes, 1957a,b). It turns out that the energy-based model is the only distribution with maximum entropy. More specifically, the following theorem holds:

**Theorem 1.** *Let $\mathcal{P}_\mu := \{p(S) : \mathbb{E}_p[F(S)] = \mu\}$ be a set of distributions satisfying the expectation constraint $\mathbb{E}_p[F(S)] = \mu$, and $p_\lambda$ have density*

$$p_\lambda(S) = \frac{\exp(\lambda F(S))}{Z}, \quad Z := \sum_{S \subseteq V} \exp(\lambda F(S)).$$

*If $\mathbb{E}_{p_\lambda}[F(S)] = \mu$, then $p_\lambda$ maximizes the entropy $\mathbb{H}(p)$ over $\mathcal{P}_\mu$; moreover, the distribution $p_\lambda$ is unique.*

*Proof.* The derivation below is adapted from Jaynes (1957a) in the context of set function learning, for completeness. We rewrite the maximum entropy problem in the form of

$$\text{maximize} - \sum_{S \subseteq V} p(S) \log p(S)$$

$$\text{subject to} \sum_{S \subseteq V} p(S)F(S) = \mu, \quad \forall S \subseteq V p(S) \geq 0, \quad \sum_{S \subseteq V} p(S) = 1.$$

Introducing Lagrange multipliers $\alpha(S) > 0$ for the constraint $p(S) > 0$, $\beta \in \mathbb{R}$ for the normalization constraint that $\sum_{S \subseteq V} p(S) = 1$, $\lambda$ for the constraint that $\mathbb{E}_p[F(S)] = \mu$, and , we obtain the following Lagrangian:

$$L(p, \alpha_0, \alpha_1, \beta) = \sum_{S \subseteq V} p(S) \log p(S) + \beta \left( \sum_{S \subseteq V} p(S) - 1 \right) +$$

$$\lambda \left( \mu - \sum_{S \subseteq V} p(S)F(S) \right) - \sum_{S \subseteq V} \alpha(S)p(S). \tag{15}$$

Now we take derivatives and obtain

$$\frac{\partial}{\partial p(S)} L(p, \alpha, \beta, \lambda) = 1 + \log p(S) + \beta - \lambda F(S) - \alpha(S). \tag{16}$$

Since this function is convex in $p$, the minimizing $p$ can be find by setting this equal to zero

$$p(S) = \exp(\lambda F(S) - 1 - \beta + \alpha(S)). \tag{17}$$

Note that in this setting we always have $p(S) > 0$. By complementary slackness, the constraint $p(S) > 0$ is unnecessary and we have $\alpha(S) = 0$. To satisfy the constraint $\sum_{S \subseteq V} p(S) = 1$, we take $\beta = 1 - + \log Z = -1 + \log \sum_{S \subseteq V} \exp(\lambda F(S))$. Then the optimal mass $p$ has the form

$$p_\lambda(S) = \frac{\exp(\lambda F(S))}{\sum_{S \subseteq V} \exp(\lambda F(S))}. \tag{18}$$

So we reach the form of $p(S)$ we would like to have.

Next we show the distribution $p_\lambda$ is unique. Assume there exists any other distribution $p \in \mathcal{P}_\mu$, such that $p = \operatorname{argmax}_p \mathbb{H}(p)$. In this case, we have

$$
\begin{aligned}
\mathbb{H}(p) &= -\sum_{S \subseteq V} p(S) \log p(S) = -\sum_{S \subseteq V} p(S) \log \frac{p(S)}{p_\lambda(S)} - \sum_{S \subseteq V} p(S) \log p_\lambda(S) \\
&= -\mathbb{KL}(p \| p_\lambda) - \sum_{S \subseteq V} p(S) \left( \lambda F(S) - Z \right) \\
&= -\mathbb{KL}(p \| p_\lambda) - \sum_{S \subseteq V} p_\lambda(S) \left( \lambda F(S) - Z \right) \\
&= -\mathbb{KL}(p \| p_\lambda) + \mathbb{H}(p_\lambda).
\end{aligned}
$$

As $\mathbb{KL}(p \| p_\lambda) \geq 0$ unless $p = p_\lambda$, we have shown that $p_\lambda$ is the unique distribution maximizing the entropy, as desired. $\qquad\square$

**Discussion.** Theorem 1 shows that EBM is the maximum entropy distribution, which verifies the assertion that energy-based treatments of set function enjoy the *minimum prior* property. It should be noted that the model proposed by Tschiatschek et al. (2018) violates this requirement. They used sequential modeling to construct $p(S)$ (see (13) and (14)). Although this approach simplifies the sampling process, it introduces undesirable inductive bias.

### B.2 Derivations of the Fixed Point Iteration

In this section, we give the detailed derivation for the fixed point iteration (FPI) of MFVI:

$$\psi_i^{(k+1)} \leftarrow (1 + \exp(-\nabla_{\psi_i^{(k)}} f_{\mathrm{mt}}^{F_\theta}(\boldsymbol{\psi}^{(k)})))^{-1}. \tag{19}$$

First, recall that we want to maximize the ELBO:

$$\max_{\boldsymbol{\psi}} \underbrace{\sum_{S \subseteq V} F_\theta(S) \prod_{i \in S} \psi_i \prod_{i \notin S} (1 - \psi_i)}_{f_{\mathrm{mt}}^{F_\theta}(\boldsymbol{\psi})} - \underbrace{\sum_{i=1}^{|V|} [\psi_i \log \psi_i + (1 - \psi_i) \log(1 - \psi_i)]}_{-\mathbb{H}(q(S; \boldsymbol{\psi}))}. \tag{20}$$

The formula (19) is obtained by setting the partial derivative *w.r.t.* coordinate $i$ of ELBO to be 0:

$$\nabla_{\psi_i} f_{\mathrm{mt}}^{F_\theta}(\boldsymbol{\psi}) + \nabla_{\psi_i} \mathbb{H}(q(S; \boldsymbol{\psi})) = \nabla_{\psi_i} f_{\mathrm{mt}}^{F_\theta}(\boldsymbol{\psi}) + \log \frac{1 - \psi_i}{\psi_i} = 0,$$

which implies

$$\psi_i = (1 + \exp(-\nabla_{\psi_i} f_{\mathrm{mt}}^{F_\theta}(\boldsymbol{\psi})))^{-1}.$$

This is exactly the formula of FPI used in the mean-field variational inference algorithm. Note that the FPI actually corresponds to the gradient ascent with an adaptive step size vector $\boldsymbol{\alpha}$ as

$$\alpha_i = \frac{\sigma(\nabla_{\psi_i} f_{\mathrm{mt}}^{F_\theta}(\boldsymbol{\psi})) - \psi_i}{\nabla_{\psi_i} f_{\mathrm{mt}}^{F_\theta}(\boldsymbol{\psi}) + \log[(1 - \psi_i)/\psi_i]},$$

where $\sigma(x) = (1 + \exp(-x))^{-1}$ denotes the sigmoid function. To verify this, we have

$$
\begin{aligned}
\psi_i^{(k+1)} &= \psi_i^{(k)} + \alpha_i^{(k)} \left( \nabla_{\psi_i^{(k)}} f_{\mathrm{mt}}^{F_\theta}(\boldsymbol{\psi}^{(k)}) + \nabla_{\psi_i^{(k)}} \mathbb{H}(q(S; \boldsymbol{\psi}^{(k)})) \right) \\
&= \psi_i^{(k)} + \frac{\sigma(\nabla_{\psi_i} f_{\mathrm{mt}}^{F_\theta}(\boldsymbol{\psi}^{(k)})) - \psi_i^{(k)}}{\nabla_{\psi_i^{(k)}} f_{\mathrm{mt}}^{F_\theta}(\boldsymbol{\psi}^{(k)}) + \log[(1 - \psi_i^{(k)})/\psi_i^{(k)}]} \left( \nabla_{\psi_i^{(k)}} f_{\mathrm{mt}}^{F_\theta}(\boldsymbol{\psi}^{(k)}) + \log \frac{(1 - \psi_i^{(k)})}{\psi_i^{(k)}} \right) \\
&= (1 + \exp(-\nabla_{\psi_i^{(k)}} f_{\mathrm{mt}}^{F_\theta}(\boldsymbol{\psi}^{(k)})))^{-1}.
\end{aligned}
$$

The connection to gradient ascent further confirms the soundness of our FPI algorithm.

### B.3    Derivations of the Gradient of Multilinear Extension

In this section, we prove that the gradient of multilinear extension can be estimated using Monte Carlo sampling. Specifically we have

$$
\begin{aligned}
\nabla_{\psi_i} f_{\mathrm{mt}}^{F_\theta} &= \nabla_{\psi_i} \sum_{S \subseteq V} F_\theta(S) \prod_{i \in S} \psi_i \prod_{i \notin S} (1 - \psi_i) \\
&= \mathbb{E}_{q(S;(\boldsymbol{\psi}|\psi_i \leftarrow 1))}[F_\theta(S)] - \mathbb{E}_{q(S;(\boldsymbol{\psi}|\psi_i \leftarrow 0))}[F_\theta(S)] \\
&= \sum_{S \subseteq V, i \in S} F_\theta(S) \prod_{j \in S \setminus \{i\}} \psi_j \prod_{j' \notin S} (1 - \psi_{j'}) - \sum_{S \subseteq V \setminus \{i\}} F_\theta(S) \prod_{j \in S} \psi_j \prod_{j' \in S, j' \neq i} (1 - \psi_{j'}) \\
&= \sum_{S \subseteq V \setminus \{i\}} [F_\theta(S + i) - F_\theta(S)] \prod_{j \in S} \psi_j \prod_{j' \in V \setminus S \setminus \{i\}} (1 - \psi_{j'}) \\
&= \mathbb{E}_{q(S;(\boldsymbol{\psi}|\psi_i \leftarrow 0))} [F_\theta(S + i) - F_\theta(S)] .
\end{aligned}
\tag{21}
$$

**Discussion.** The Monte Carlo (MC) approximation of $\nabla_{\psi_i} f_{\mathrm{mt}}^{F_\theta}$ is unbiased. Thereby, although exactly calculating (21) has exponential time complexity, we can apply MC sampling to approximate it in a polynomial time, resulting a scalable training algorithm. It is worth to note that the MC approximation used in PGM (see (14)) is biased. That is they approximate $\log p_\theta(S|V) = \log \sum_{\pi \in \Pi^S} p_\theta(\pi|V)$ with $\log p_\theta(\pi|V), \pi \sim \Pi^S$. Although such a biased approximation can be computed in polynomial time, they undesirably introduce permutation variance.

## C    Low-Rank Perturbation for the Covariance Matrix

In the construction of Gaussian copula $C_{\boldsymbol{\Sigma}}$, we require a positive semi-definite matrix $\boldsymbol{\Sigma} \in \mathbb{R}^{|V| \times |V|}$, whose elements are generally modeled as the output of neural networks. Thereby, if the size of ground size $V$ is large, the number of neural network outputs will be prohibitively large. Meanwhile, based on the definition of set, covariance matrix $\boldsymbol{\Sigma}$ is further required to satisfy *permutation equivariance*. To remedy this issue, we propose to employ a more efficient strategy, namely *Lower-Rank Perturbation*, which restricts the covariance matrix to the form

$$
\boldsymbol{\Sigma} = \boldsymbol{D} + \boldsymbol{P}\boldsymbol{P}^T,
\tag{22}
$$

where $\boldsymbol{\Sigma} \in \mathbb{R}_+^{|V| \times |V|}$ is a diagonal matrix with positive entries and $\boldsymbol{P} = [\boldsymbol{p}_1, \boldsymbol{p}_2, \ldots, \boldsymbol{p}_v]$ is a lower-rank perturbation matrix with $\boldsymbol{p}_i \in \mathbb{R}^{|V|}$ and $v \ll |V|$. In this way, the number of neural network outputs can be dramatically reduced from $|V|^2$ to $v|V|$. Another benefit of constructing $\boldsymbol{\Sigma}$ in this way is that, it is convenient to employ the DeepSet architecture in (12) to output $\boldsymbol{D}$ and $\boldsymbol{p}_i$ for $i = 1, \ldots, v$, such that they are *permutation equivariant*, and the resulting covariance matrix $\boldsymbol{\Sigma} = \boldsymbol{D} + \boldsymbol{P}\boldsymbol{P}^T$ is also *permutation equivariant*. Moreover, the lower-rank perturbation trick permits us to avoid using Cholesky decomposition to sample a Gaussian noise with covariance $\boldsymbol{\Sigma}$, which is prohibitively expensive. Specifically, the Gaussian noise $\boldsymbol{g} \sim \mathcal{N}(\boldsymbol{0}, \boldsymbol{\Sigma})$ can be reparameterized as

$$
\boldsymbol{g} = \boldsymbol{D}^{1/2} \cdot \boldsymbol{\epsilon}_1 + \boldsymbol{P} \cdot \boldsymbol{\epsilon}_2,
\tag{23}
$$

where $\boldsymbol{\epsilon}_1 \sim \mathcal{N}(\boldsymbol{0}, \boldsymbol{I}_{|V|})$ and $\boldsymbol{\epsilon}_2 \sim \mathcal{N}(\boldsymbol{0}, \boldsymbol{I}_v)$. In this way, the sampling complexity can be reduced from $\mathcal{O}(|V|^3)$ to $\mathcal{O}(v^2|V|)$.

## D    Detailed Pseudo Code of EquiVSet Algorithms

We provide the pseudo-code for `EquiVSet` in Algorithm 4. The training procedure consists of two steps: i) train $q_\phi$ with fixed $\theta$; ii) train $p_\theta$ under the guidance of $q_\phi$. Specifically, to train $q_\phi$, we first fix the parameter $\theta$ of the set function and then optimize $\phi$ by maximizing the ELBO in (6). To train $p_\theta$, we first initialize the variational parameter $\boldsymbol{\psi}$ via EquiNet and then run $K$ steps mean-field iteration to make $\boldsymbol{\psi}$ dependent with $\theta$. Finally, the parameter $\theta$ can be optimized by minimizing the cross entropy in (5). Note that, we set $K$ as 1 in our experiments.

**Algorithm 4** EquiVSet (complete version)

**Input**: $\{V_i, S_i^*\}_{i=1}^N$: training dataset; $\eta$: learning rate; $K$ : number of mean-field iteration step; $m$ : number of Monte Carlo approximations; $v$: rank of perturbation; $\tau$: temperature for Gumbel-Softmax
**Output**: Optimal parameters $(\theta, \phi)$

1: $\theta, \phi \leftarrow$ Initialize parameter
2: **repeat**
3:     Sample training data point
    $(V, S^*) \sim \{V_i, S_i^*\}_{i=1}^N$
4:     Obtain variational parameter $\boldsymbol{\psi}$ via EquiNet
    $\boldsymbol{\psi} \leftarrow \mathrm{EquiNet}(V; \phi)$
5:     Sample $m$ subsets via $\mathrm{CopulaBernoulli}(V, v, \tau)$ or $\mathrm{IndBernoulli}(V, \tau)$
    $S_n \sim q(S; \boldsymbol{\psi}), n = 1, 2, \ldots, m$
6:     Update the parameter $\phi$ by maximizing ELBO in (6)     **Optimize $\phi$**

$$\phi \leftarrow \phi + \eta \nabla_\phi \left( \frac{1}{m} \sum_{n=1}^m F_\theta(S_n) - \sum_{i=1}^{|V|} [\psi_i \log \psi_i + (1-\psi_i)\log(1-\psi_i)] \right)$$

7:     Initialize parameter $\boldsymbol{\psi}$ via EquiNet
    $\boldsymbol{\psi}^{(0)} \leftarrow \mathrm{EquiNet}(\mathrm{V}; \mathrm{stop\_gradient}(\phi))$
8:     **for** $k \leftarrow 1, \ldots, K$ **do**
9:       **for** $i \leftarrow 1, \ldots, |V|$ in parallel **do**
10:         Sample $m$ subsets via the variational distribution[10]     **Mean-field Iteration**
        $S_n \sim q(S; \boldsymbol{\psi}^{(0)}|\psi_i^{(0)} \leftarrow 0), n = 1, 2, \ldots, m$    $\mathrm{MFVI}(\boldsymbol{\psi}^{(0)}, V, K)$
11:         Update the variational parameter $\boldsymbol{\psi}$

$$\psi_i^* \leftarrow \sigma \left( \frac{1}{m} \sum_{n=1}^m [F_\theta(S_n + i) - F_\theta(S_n)] \right)$$

12:       **end for**
13:     **end for**
14:     Update the parameter $\theta$ by minimizing the cross entropy loss in (5)     **Optimize $\theta$**

$$\theta \leftarrow \theta - \eta \nabla_\theta \left( -\sum_{i \in S^*} \log \psi_i^* - \sum_{i \in V \setminus S^*} \log(1 - \psi_i^*) \right)$$

15: **until** convergence of parameters $(\boldsymbol{\theta}, \boldsymbol{\phi})$

---

**Algorithm 5** $\mathrm{IndBernoulli}(V, \tau)$

**Input**: $V$: ground set; $\tau$: temperature for Gumbel-Softmax
**Output**: Sampled subset $\boldsymbol{s}$

1: Obtain location parameter $\boldsymbol{\psi}$ via EquiNet: $\boldsymbol{\psi} \leftarrow \mathrm{EquiNet}(V; \phi)$
2: Draw uniform noise: $u_i \sim \mathcal{U}(0, 1), i = 1, \ldots, |V|$
3: Apply Gumbel-Softmax trick: $\tilde{s}_i = \sigma \left( \frac{1}{\tau} \left( \log \frac{\psi_i}{1-\psi_i} + \log \frac{u_i}{1-u_i} \right) \right), i = 1, \ldots, |V|$
4: Apply Straight-Through estimator: $\boldsymbol{s} = \mathrm{stop\_gradient}(\mathbb{I}(\tilde{\boldsymbol{s}} \geq \boldsymbol{\epsilon}) - \tilde{\boldsymbol{s}}) + \tilde{\boldsymbol{s}}, \boldsymbol{\epsilon} \sim \mathcal{U}(\boldsymbol{0}, \boldsymbol{I})$

---

**Algorithm 6** $\mathrm{CopulaBernoulli}(V, v, \tau)$

**Input**: $V$: ground set; $v$: rank of perturbation; $\tau$: temperature for Gumbel-Softmax
**Output**: Sampled subset $\boldsymbol{s}$

1: Obtain location parameter $\boldsymbol{\psi}$ via EquiNet: $\boldsymbol{\psi} \leftarrow \mathrm{EquiNet}(V; \phi)$
2: Draw Gaussian noise:
   { In the following, $\boldsymbol{D}$ is a diagonal matrix and $\boldsymbol{P}$ is the lower-rank perturbation matrix. }
   $\boldsymbol{g} = \boldsymbol{D}^{1/2} \cdot \boldsymbol{\epsilon}_1 + \boldsymbol{P} \cdot \boldsymbol{\epsilon}_2, \boldsymbol{P} = [\boldsymbol{p}_1, \boldsymbol{p}_2, \ldots, \boldsymbol{p}_v], \boldsymbol{\epsilon}_1 \sim \mathcal{N}(\boldsymbol{0}, \boldsymbol{I}_{|V|})$ and $\boldsymbol{\epsilon}_2 \sim \mathcal{N}(\boldsymbol{0}, \boldsymbol{I}_v)$
3: Apply element-wise Gaussian CDF: $\boldsymbol{u} = \boldsymbol{\Phi}_{\mathrm{diag}(\boldsymbol{D}+\boldsymbol{P}\boldsymbol{P}^T)}(\boldsymbol{g})$
4: Apply Gumbel-Softmax trick: $\tilde{s}_i = \sigma \left( \frac{1}{\tau} \left( \log \frac{\psi_i}{1-\psi_i} + \log \frac{u_i}{1-u_i} \right) \right), i = 1, \ldots, |V|$
5: Apply Straight-Through estimator: $\boldsymbol{s} = \mathrm{stop\_gradient}(\mathbb{I}(\tilde{\boldsymbol{s}} \geq \boldsymbol{\epsilon}) - \tilde{\boldsymbol{s}}) + \tilde{\boldsymbol{s}}, \boldsymbol{\epsilon} \sim \mathcal{U}(\boldsymbol{0}, \boldsymbol{I})$

---

[10]Here we apply mean-field variational inference, which means the variational distribution $q$ is an independent Bernoulli distribution.

# E   Experimental Details

## E.1   The Architecture of EquiVSet

In this section, we provide a detail architecture description of $\texttt{EquiVSet}_{\text{copula}}$. $\texttt{EquiVSet}_{\text{copula}}$ consists of two different components that are implemented as neural networks: (i) the set function which is permutation invaraint and (ii) the recognition network which is permutation equivariant. We employ the DeepSet architecture to implement these two components, with the detailed architectures are given in Table 5.

Table 5: Detailed architectures of $\texttt{EquiVSet}_{\text{copula}}$.

| Set Function | Recognition Network | |
|---|---|---|
| $\text{InitLayer}(S, 256)$ | $\text{InitLayer}(V, 256)$ | |
| $\text{SumPooling}$ | $\text{FC}(256, 500, \text{ReLU})$ | |
| $\text{FC}(256, 500, \text{ReLU})$ | $\text{FC}(500, 500, \text{ReLU})$ | |
| $\text{FC}(500, 500, \text{ReLU})$ | | $\boldsymbol{D} = \text{diag}(\text{FC}(500, 1, \text{softplus}))$ |
| $\text{FC}(256, 1, -)$ | $\boldsymbol{\psi} = \text{FC}(500, 1, \text{sigmoid})$ | $\boldsymbol{P} = [\text{FC}(500, 1, \text{tanh})]^{v}$ |

In Table 5, $\text{InitLayer}(S, d)$ denotes the set transformation function, which encodes the set objects into vector representations. $\text{FC}(d, h, f)$ denotes the fully-connected layer with activation function $f$. $\text{diag}(\boldsymbol{v})$ is a diagonal matrix with the elements of diagonal being vector $\boldsymbol{v}$. $[\boldsymbol{p}]^v = [\boldsymbol{p}_1, \ldots, \boldsymbol{p}_v]$ denotes a matrix with $\boldsymbol{p}$ representing a column perturbation vector. Note that we also propose two variant methods, $i.e.,$ $\texttt{DiffMF}$ and $\texttt{EquiVSet}_{\text{ind}}$. For $\texttt{DiffMF}$, we apply the same architecture of the the set function in Table 5. We also exploit the same architecture for $\texttt{EquiVSet}_{\text{ind}}$, but discarding the copula components, $i.e.,$ $\boldsymbol{D}$ and $\boldsymbol{P}$. In all experiments, we implement our models following the same architecture with the difference being that we apply various $\text{InitLayer}$ to different datasets. The architectures of $\text{InitLayer}$ for different datasets are depicted below.

**Synthetic datasets.** The synthetic datasets consist of the Tow-Moons and Gaussian-Mixture datasets. Each instance of the set is a two-dimensional vector, which represents the corresponding Cartesian coordinates. In this dataset, the $\text{InitLayer}$ is a one-layer feed-forward neural network $\text{FC}(2, 256, -)$.

**Amazon Baby Registry.** The Amazon baby registry dataset consists of a set of products that are characterized by a short textual description. We transform them into vector representations using the pre-trained BERT module (Devlin et al., 2018). Thereby, each instance of the set is a 768 dimensional feature vector. The $\text{InitLayer}$ is modelled as $\text{FC}(768, 256, -)$.

**Double MNIST.** The double MNIST dataset consists of different digit images ranging from $00$ to $99$. Each image has the shape of $(64, 64)$ and we reshape it into $(4096, )$. Therefore, the $\text{InitLayer}$ is designed as $\text{FC}(4096, 256, -)$.

**CelebA.** The CelebA dataset contains $202,599$ number of face images. Each image is in the shape of $(3, 64, 64)$. We employ convolutional neural networks as $\text{InitLayer}$. Specifically, the architecture of $\text{InitLayer}$ is

$$\text{ModuleList}([\text{Conv}(32, 3, 2, \text{ReLU}), \text{Conv}(64, 4, 2, \text{ReLU}),$$
$$\text{Conv}(128, 5, 2, \text{ReLU}), \text{MaxPooling}, \text{FC}(128, 256, -)]),$$

where $\text{Conv}(d, k, s, f)$ is a convolutional layer with $d$ output channels, $k$ kernel size, $s$ stride size, and activation function $f$.

**PDBBind.** The PDBBind database consists of experimentally measured binding affinities for biomolecular complexes (Liu et al., 2015a). It provides detailed 3D Cartesian coordinates of both ligands and their target proteins derived from experimental (*e.g.*, X-ray crystallography) measurements. The atomic convolutional network (ACNN) (Gomes et al., 2017) provides meaningful vector features for complexes by constructing nearest neighbor graphs based on the 3D coordinates of atoms and predicting binding free energies. In this work, we apply the output of last second layer of the ACNN model followed by feed-forward neural networks to obtain the representations of complexes. More formally, the $\text{InitLayer}$ is defined as

$$\text{ModuleList}([\text{ACNN}[: -1], \text{FC}(1922, 2048, \text{ReLU}), \text{FC}(2048, 256, -)]),$$

where $\mathrm{ACNN}[:-1]$ denotes the ACNN module without the last prediction layer, whose output dimensionality is 1922.

**BindingDB.** The BindingBD dataset contains $52,273$ drug-target pairs. We exploit the DeepDTA model (Öztürk et al., 2018) to encode drug-target pairs as vector representations. Specifically, the DeepDTA model first represents the drug compound and target protein as sequences of one-hot vectors and encodes them as feature vectors using convolutional neural networks. The detailed architecture of InitLayer used in this dataset is demonstrated in Table 6.

Table 6: Detailed architectures of InitLayer in the BindingDB dataset.

| Drug | Target |
|---|---|
| $\mathrm{Conv}(32, 4, 1, \mathrm{ReLU})$ | $\mathrm{Conv}(32, 4, 1, \mathrm{ReLU})$ |
| $\mathrm{Conv}(64, 6, 1, \mathrm{ReLU})$ | $\mathrm{Conv}(64, 8, 1, \mathrm{ReLU})$ |
| $\mathrm{Conv}(96, 8, 1, \mathrm{ReLU})$ | $\mathrm{Conv}(96, 12, 1, \mathrm{ReLU})$ |
| MaxPooling | MaxPooling |
| $\mathrm{FC}(96, 256, \mathrm{ReLU})$ | $\mathrm{FC}(96, 256, \mathrm{ReLU})$ |
| Concat | |
| $\mathrm{FC}(512, 256, -)$ | |

## E.2 Implementation Details

Here we provide a detailed description of the hyperparameters setup for our model `EquiVSet` and its variants. `EquiVSet` contains four important hyperprameters: the number of Monte Carlo sampling $m$ and mean-field iteration steps $K$ in Algorithm 1, and the rank of lower-rank perturbation $v$ in (22). We set $m = 5, v = 5$ throughout the experiments. For the mean-field iteration steps $K$, we set it as 5 for the variant model `DiffMF`, and 1 for `EquiVSet`$_{\mathrm{ind}}$ and `EquiVSet`$_{\mathrm{copula}}$. It is noted that the hyperparameters above are empirically set, and we have detail sensitivity analysis in Appendix F.5. The proposed models are trained using the Adam optimizer (Kingma & Ba, 2014) with a fixed learning rate $1e-4$ and weight decay rate $1e-5$. We choose the batch size from $\{4, 8, 16, 32, 64, 128\}$, since the model sizes for various datasets are different and we choose the largest batch size to enable it can be trained on a single Tesla V100-SXM2-32GB GPU.

We apply the early stopping strategy to train the models, including the baselines and our models. That is if the performances are not improved in continuous 6 epochs, we early stop the training process. Each dataset is trained for maximum 100 epochs. After each epoch, we validate the model and save the model with the best performance on the validation set. After training, we evaluate the performance of saved models on the test set. We repeat all experiments 5 times with different random seeds and the average performance metrics and their standard deviations are reported as the final performances.

## E.3 Baselines

Throughout the experiments, we compared our models with three conventional approaches: random guess, probabilistic greedy model (PGM) (Tschiatschek et al., 2018) and DeepSet (Zaheer et al., 2017). Further descriptions of the benchmarks and implementation details are as follows.

- Random: We report the expected value of the Jaccard coefficient (JC) of random guess. This baseline provides an estimate of how difficult the task is. Specifically, given a data point $(V, S^*)$, it can be computed as $\mathbb{E}(JC(V, S^*)) = \sum_{k=0}^{|S^*|} \frac{\binom{|S^*|}{k}\binom{|V|-|S^*|}{|S^*|-k}}{\binom{|V|}{|S^*|}} \frac{k}{2|S^*|-k}$.

- PGM (Tschiatschek et al., 2018): PGM is the most relevant method to the set functions learning under the OS oracle, that solves optimization (1) using greedy maximization algorithm with the virtues of differentiability and permutation invariance. We employ the same architecture defined in Table 5 to model the set function $F_\theta(S)$ in (13). The temperature parameter $\gamma$ is empirically set as 1. We use Monte Carlo sampling to estimate (14). That is we randomly sample one permutation $\pi \sim \Pi^S$ and use $p_\theta(\pi|V)$ to approximate $p_\theta(S|V)$. The model is trained using the Adam optimizer, with batch size choosing from $\{4, 8, 16, 32, 64, 128\}$, fixed learning rate $1e-4$, and fixed weight decay rate $1e-5$.

Table 7: The statistics of Amazon product dataset. #products: number of all products in the category. $|\mathcal{D}|$: number of samples in the dataset.

| Categories | #products | $|\mathcal{D}|$ | $|V|$ | $\sum |S^*|$ | $\mathbb{E}[|S^*|]$ | $\min_{S^*} |S^*|$ | $\max_{S^*} |S^*|$ |
|---|---|---|---|---|---|---|---|
| Toys | 62 | 2,421 | 30 | 9,924 | 4.09 | 3 | 14 |
| Furniture | 32 | 280 | 30 | 892 | 3.18 | 3 | 6 |
| Gear | 100 | 4,277 | 30 | 16,288 | 3.80 | 3 | 10 |
| Carseats | 34 | 483 | 30 | 1,576 | 3.26 | 3 | 6 |
| Bath | 100 | 3,195 | 30 | 12,147 | 3.80 | 3 | 11 |
| Health | 62 | 2,995 | 30 | 11,053 | 3.69 | 3 | 9 |
| Diaper | 100 | 6,108 | 30 | 25,333 | 4.14 | 3 | 15 |
| Bedding | 100 | 4,524 | 30 | 17,509 | 3.87 | 3 | 12 |
| Safety | 36 | 267 | 30 | 846 | 3.16 | 3 | 5 |
| Feeding | 100 | 8,202 | 30 | 37,901 | 4.62 | 3 | 23 |
| Apparel | 100 | 4,675 | 30 | 21,176 | 4.52 | 3 | 21 |
| Media | 58 | 1,485 | 30 | 6,723 | 4.52 | 3 | 19 |

- DeepSet (NoSetFn) (Zaheer et al., 2017): DeepSet is a neural-network-based architecture that satisfies permutation invariance and varying ground sets. Although the DeepSet architecture can be employed here to sample the optimal subset oracle, it does not learn the set functions explicitly. We exploit the same architecture of $F_\theta(S)$ in Table 5, but drop the SumPooling operator to ensure the dimensionality of output is $|V|$. This baseline is trained by minimizing the objective in (5) using the Adam optimizer with batch size choosing from $\{4, 8, 16, 32, 64, 128\}$, fixed learning rate $1e - 4$, and fixed weight decay rate $1e - 5$.

### E.4 Assumptions on the Underlying Data Generative Distribution of the OS Oracle

In this section, we discuss the assumptions made about the data distribution for better understanding the set functions learning under optimal subset (OS) oracle. Generally speaking, for any scenario with the output being a subset $S$ of the given ground set $V$ of the input, the proposed approach could be applied to predict the subset $S$ of the given ground set $V$. The only loose assumption is that the optimal subset oracle $S^*$ of a given ground set $V$ is generated by some underlying distribution formulated via a utility function that maximizes the utility value of OS oracle (see (1) in the main text). We further assume the utility function could be parameterized by a deep neural network, thanks to the universal approximation theorem (Leshno et al., 1993).

This assumption is very weak and generally makes sense in practice. We also apply this assumption to the datasets used in the experiments. Specifically, in the product recommendation (Appendix E.5), $V$ is the set of recommended products, and $S^*$ is the one the customer buys (or adds to the cart). Undoubtedly, the underlying generative distribution, or say the utility function is specified by the selection process of customers. In the set anomaly detection (Appendix E.6), given a ground set $V$, $S^*$ is generated as the one containing anomaly data points. Therefore, the utility function in this setting is formulated as the anomaly pattern. Moreover, in the compound selection Appendix E.7, we applied high bioactivity and diversity filters to select compounds. In this case, the utility function is determined by the bioactivity and diversity of the group of compounds.

### E.5 Detailed Experimental Settings for Product Recommendation

**Detailed Descriptions of the Amazon Baby Registry Dataset.** The Amazon baby registry data (Gillenwater et al., 2014) consists of baby registry data collected from Amazon and is split into several datasets according to product categories, such as toys, furniture, etc. For each category, which can be considered as the product database, Amazon provides multiple sets of products selected by different customers. Thereby, these subsets of products can be viewed as OS oracles. To ensure that each ground set $V$ only contains one OS oracle $S^*$, we construct the sample $(V, S^*)$ as follows. For each subset of products selected by an anonymous user, we filter it out if its size is equal to 1 or larger than 30. For each OS oracle $S^*$ in the remaining subsets, we randomly sample $30 - |S^*|$ products in the same category to construct $V \backslash S^*$. We summarize the statistics of the categories in Table 7.

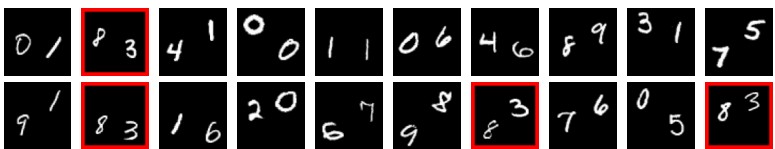

Figure 4: A sampled data for the Double MNIST dataset, which consists of $|S^*|$ images with the same digit (red box, 83 in this case) and $20 - |S^*|$ images with different digits.

**Comparing with the Setting of (Tschiatschek et al., 2018, Section 5.3).** In (Tschiatschek et al., 2018, Section 5.3), Tschiatschek et al. (2018) consider an alternative setting which is different from ours. Specifically, they construct the ground set $V$ as all the products in a category, and view the selected subsets of all the customers as the corresponding optimal subsets. That is why they have the data points in the form of $\{V, (S_1^*, \ldots, S_N^*)\}$. This is a bit problematic since it is deviated from the real world scenario: naturally the chosen subset $S_i^*$ shall depend on both $V$ and the $i$-th customer's personal preference. However, the customer is fully anonymized, so no information can be extracted from this dataset.

In order to be aligned with the real world scenario, we curate the dataset in the following way, in order to make data samples in the OS supervision oracle with the data in the form of $\{V_i, S_i^*\}$.

For each category, we generate samples $(V, S^*)$ as follows. Firstly, we filter out those subsets selected by customers whose size is equal to 1 or larger than 30. Then we split the remaining subset collection $\mathcal{S}$ into training, validation and test folds with a $1 : 1 : 1$ ratio. Finally for each OS oracle $S^* \in \mathcal{S}$, we randomly sample additional $30 - |S^*|$ products from the same category to construct $V \backslash S^*$.

In this way, we construct one data point $(V, S^*)$ for each customer, which reflects this real world scenario: $V$ contains 30 products displayed to the customer, and the customer is interested in checking $|S^*|$ of them. This is also consistent with real world recommender system, as users can only browse a small number of products at a time since the screen size of the device is limited, and the user has limited attention.

### E.6  Detailed Experimental Settings for Set Anomaly Detection

In this experiment, we evaluate our methods on two real-world datasets:

**Double MNIST:** The dataset consists of 1000 images for each digit ranging from 00 to 99. For each sample $(V, S^*)$, we randomly sample $n \in \{2, \ldots, 5\}$ images with the same digit to construct the OS oracle $S^*$, and then select $20 - |S^*|$ images with different digits to construct the set $V \backslash S^*$. An example is shown in Figure 4.

**CelebA:** The CelebA dataset contains $202,599$ images with 40 attributes. As shown in Figure 5, we select two attributes at random and construct the set with the size of 8. For each ground set $V$, we randomly select $n \in \{2, 3\}$ images as the OS oracle $S^*$, in which neither of the two attributes is present. In this way, we arrive at train, val, test datasets with 10,000, 1000, 1000 samples respectively.

### E.7  Detailed Experimental Settings for Compound Selection

Algorithm 7 shows the corresponding data generation process of simulating the OS oracle for compound selection. In this algorithm, $\mathrm{random\_choose}(\mathcal{C}, n)$ means randomly choosing $n$ compounds from the database $\mathcal{C}$ (*i.e.,* PDBBind or BindingDB), and $\mathrm{topK\_bioactivity}(V, m)$ represents selecting the top-$m$ compounds with highest biological activity from the ground set $V$. These two operators combine together to form the bioactivity filter, in which we set $(n, m)$ as $(30, 10)$, and $(300, 100)$ for PDBBind and BindingDB, respectively. To further apply the diversity filter, we use the RDKit[11] tools to compute the similarity between each molecule pair based on their topological fingerprints. This operator corresponds to the line 3 of Algorithm 7, in which $\mathrm{cal\_fingerprint\_similarity}(S)$ returns the similarity matrix $M \in \mathbb{R}^{|S| \times |S|}$ of the set of compounds $S$. Since rows (or columns) of the similarity matrix can be regarded as the features of the corresponding molecules, the molecules are clustered based on these similarity features by applying the affinity propagation algorithm. The OS

---

[11] https://github.com/rdkit/rdkit

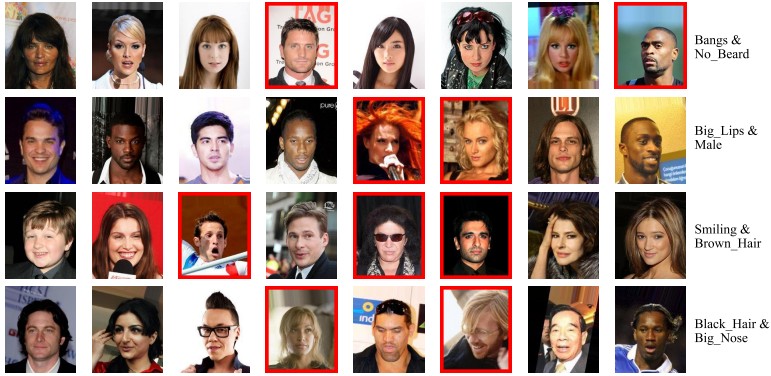

Figure 5: Sampled data points for the CelebA dataset. Each row is a sample, consisting of $|S^*|$ anomalies (red box) and $8 - |S^*|$ normal images. In each sample, a normal image has two attributes (rightmost column) while anomalies do not have neither of them.

---

**Algorithm 7** OS Oracle Generation Algorithm for Compound Selection Task

---

**Input**: $\mathcal{C}$: compound database; $n$: size of ground set; $m$: number of the most active compounds
**Output**: Data point $(V, S^*)$

1: Randomly select $n$ compounds to construct the ground set
   $V \leftarrow \text{random\_choose}(\mathcal{C}, n)$
2: Filter out $m$ compounds with the highest bioactivity    **bioactivity filter**
   $S \leftarrow \text{topK\_bioactivity}(V, m)$
3: Calculate the similarity matrix
   $M \leftarrow \text{cal\_fingerprint\_similarity}(S)$
4: Apply the affinity propagation algorithm    **diversity filter**
   $\text{af} \leftarrow \text{affinity\_propogation}(M)$
5: Assign the OS oracle as cluster centers
   $S^* \leftarrow \text{af.cluster\_centers\_indices}$

---

oracle $S^*$ is finally represented by the center of each cluster. Note that, each compound consists of two small molecules, *i.e.,* the protein-ligand molecules in PDBBind, and the drug-target molecules in BindingDB. We use the protein and drug molecules to compute the fingerprint similarity for PDBBind and BindingDB, respectively.

# F  Additional Experiments

## F.1  Experiments on Varying Ground Set

Thanks to the virtues of DeepSet, our models are able to process input sets of variable sizes, which is termed as *varying ground set* property. To examine the impact of ground set sizes, we care about the following two questions: i) how well the model performs on different sizes of ground set during the test time; and ii) how well does the model train on ground sets of different sizes? To answer these two questions, we conduct experiments on the synthetic datasets using the proposed model `EquiVSet`$_{\text{copula}}$.

**Set Size Transferability Analysis**   We first experiment to understand the pattern of set size transferability. In this experiment, we train the model using fixed sizes of the ground set but test the trained model on different sizes. We present two scenarios: train on a small size but test on a large one, and train on a large size but test on a small one. For the former one, we fix the size of OS oracle $S^*$ to be 10, and train the model with ground set $V$ of size 100. After training, we test it using varying sizes of ground set in the range of $\{200, 400, 600, 800, 1000\}$. For the latter one, we fix the size of OS oracle $S^*$ to be 10, and train the model with ground set $V$ of size 1000. After training, we test it using varying sizes of ground set in the range of $\{100, 200, 400, 600, 800\}$. The former and latter experiments are conducted on the Two-Moons and Gaussian-Mixture datasets, respectively, with the

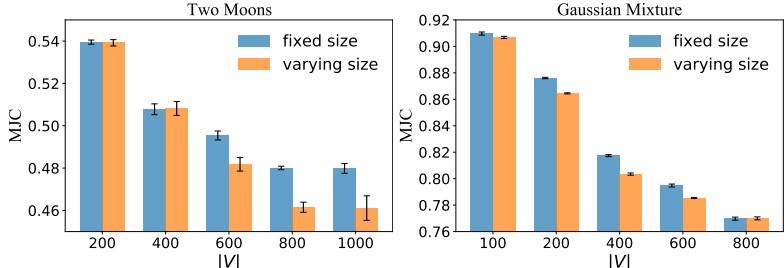

Figure 6: Synthetic results of EquiVSet$_{\texttt{copula}}$ for set size transferability analysis, in which the blue bars represent the performances of using the same sizes of ground set during the training and test time, while the yellow bars mean using different sizes of ground set during the test time. Detailed descriptions are given in the main text.

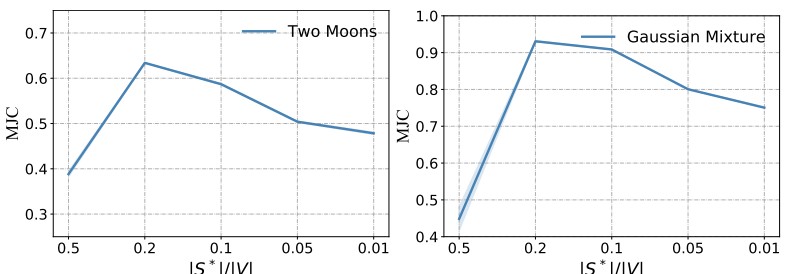

Figure 7: Synthetic results of EquiVSet$_{\texttt{copula}}$ with varying selection ratios.

results shown in Figure 6. As can be seen, the performance would be slightly reduced if tested on a different size. Moreover, increasing the difference would enlarge the reduction.

**Selection Ratio Analysis** To answer the second question, we fix the size of OS oracle $S^*$ to be 10, and experiment with different selection ratios $\frac{|S^*|}{|V|}$ in the range of $\{0.5, 0.2, 0.1, 0.05, 0.01\}$. Unlike the set size transferability analysis, in this experiment, the selection ratios are the same during training and testing. Figure 7 shows the performance of different ratios on two synthetic datasets. We observe that increasing the ratio would deteriorate the model performance. This phenomenon makes intuitive sense, since sampling subset from a large collection is more difficult. Moreover, the model performs worst when the ratio is equal to $0.5$. This is partly because $S^*$ and $V \backslash S^*$ are randomly sampled from one of two components. When $|S^*| = |V \backslash S^*|$, the model struggles to identify the optimal subset.

### F.2 Comparisons with Set Transformer

Set Transformer (Lee et al., 2019), which satisfies permutation invariant, is a well-known architecture used to model interactions among elements in the input set. Similar to DeepSet, Set Transformer could be adapted to serve as a baseline. Specifically, the architecture of the SetTransformer (NoSetFn) baseline is

$$\text{ModuleList}(\text{InitLayer}(V, 256), \text{SAB}(256, 500, 2, -), \text{SAB}(500, 1, 2, \text{sigmoid})),$$

where $\text{SAB}(d, h, m, f)$[12] denotes the set attention block (Lee et al., 2019) with $d$ dimensional set input, $h$ dimensional set output, $m$ multi-head attentions, and activation function $f$. We train the adapted Set Transformer model: $2^V \rightarrow [0, 1]^{|V|}$ with cross entropy loss and sample the subset via the topN rounding. It is noteworthy that, like DeepSet (NoSetFn), SetTransformer (NoSetFn) does not learn a set function explicitly, although it can be adapted as a baseline and can be viewed as merely modelling the amortized network in our EquiVSet framework.

---

[12]We take the implementation of SAB from https://github.com/juho-lee/set_transformer.

Table 8: Product recommendation results on Set Transformer baselines and backbones.

| Categories | Set Transformer (NoSetFn) | DiffMF (ours) | EquiVSet$_{ind}$ (ours) | EquiVSet$_{copula}$ (ours) |
|---|---|---|---|---|
| Toys | $0.640 \pm 0.030$ | $0.690 \pm 0.030$ | $0.680 \pm 0.020$ | $\mathbf{0.717 \pm 0.006}$ |
| Furniture | $\mathbf{0.175 \pm 0.008}$ | $0.170 \pm 0.020$ | $0.159 \pm 0.006$ | $0.166 \pm 0.007$ |
| Gear | $0.639 \pm 0.006$ | $0.750 \pm 0.030$ | $0.690 \pm 0.020$ | $\mathbf{0.700 \pm 0.010}$ |
| Carseats | $0.219 \pm 0.005$ | $\mathbf{0.219 \pm 0.006}$ | $0.219 \pm 0.009$ | $0.216 \pm 0.008$ |
| Bath | $0.725 \pm 0.005$ | $0.800 \pm 0.020$ | $0.800 \pm 0.010$ | $\mathbf{0.810 \pm 0.010}$ |
| Health | $0.680 \pm 0.010$ | $0.750 \pm 0.020$ | $0.750 \pm 0.020$ | $\mathbf{0.760 \pm 0.020}$ |
| Diaper | $0.789 \pm 0.005$ | $0.871 \pm 0.009$ | $0.870 \pm 0.010$ | $\mathbf{0.886 \pm 0.009}$ |
| Bedding | $0.760 \pm 0.020$ | $0.859 \pm 0.008$ | $0.860 \pm 0.020$ | $\mathbf{0.860 \pm 0.007}$ |
| Safety | $0.257 \pm 0.005$ | $0.240 \pm 0.006$ | $0.240 \pm 0.010$ | $\mathbf{0.260 \pm 0.030}$ |
| Feeding | $0.783 \pm 0.006$ | $\mathbf{0.886 \pm 0.004}$ | $0.881 \pm 0.010$ | $0.878 \pm 0.009$ |
| Apparel | $0.680 \pm 0.020$ | $0.760 \pm 0.010$ | $0.550 \pm 0.010$ | $\mathbf{0.770 \pm 0.010}$ |
| Media | $0.540 \pm 0.020$ | $0.615 \pm 0.008$ | $0.610 \pm 0.010$ | $\mathbf{0.620 \pm 0.009}$ |

Figure 8: Sampled data points for the F-MNIST (left) and CIFAR-10 (right) datasets. Each row is a sample, containing of $|S^*|$ images (red box) with the same label (rightmost column) and $8 - |S^*|$ images with different labels.

For fair comparison, we also replace the DeepSet backbone with Set Transformer in `EquiVSet`. Specifically, the InitLayer in Table 5 is replaced with

$$\mathrm{ModuleList}(\mathrm{InitLayer}(S, 256), \mathrm{SAB}(256, 500, 2, -), \mathrm{SAB}(500, 256, 2, -)).$$

Experiments are conducted on product recommendations, with the results shown in Table 8. It shows that the proposed approaches with the Set Transformer backbone outperform the Set Transformer (NoSetFn) comprehensively. One could also compare the results of Table 2 in the paper. It can be seen that the proposed `EquiVSet` (with DeepSet backbone) also performs better than the Set Transformer baseline. Moreover, `EquiVSet` (with DeepSet backbone) outperforms `EquiVSet` (with Set Transformer backbone) consistently, indicating that `EquiVSet` has great potential to be improved with more advanced architecture.

### F.3 Experiments on Set Anomaly Detection with F-MNIST and CIFAR-10

In this experiment, we further perform set anomaly detection on the other two datasets: F-MNIST (Xiao et al., 2017) and CIFAR-10 (Krizhevsky et al., 2009). Both two datasets contain images with 10 different labels. For each dataset, we randomly sample $n \in \{2, 3\}$ images as the OS oracle $S^*$, and then select $8 - |S^*|$ images with different labels to construct the set $V \backslash S^*$. We finally obtain the training, validation, and test set with the size of $10,000, 1,000, 1,000$, respectively, for both two datasets. Illustrations of sampled data are shown in Figure 8.

The results are shown in Figure 8. We see that the variants of our model consistently outperform baseline methods strongly. Moreover, `DiffMF` seems to perform better than `EquiVSet`$_{ind}$ and `EquiVSet`$_{copula}$ in set anomaly detection (similar results can be found in Table 3). However, this is not a consistent phenomenon. It seems that in most scenarios, e.g., product recommendation, compound selection, and synthetic dataset, `EquiVSet` performs better than `DiffMF`.

### F.4 Experiments on Compound Selection with Only the Bioactivity Filter

To further evaluate the potential of `EquiVSet` for drug discovery, we consider an alternative setting here. In contrast to the task in Section 6, which aims at selecting the most active compounds while

Table 9: Set anomaly detection results on the F-MNIST and CIFAR-10.

| Method | F-MNIST | CIFAR-10 |
|---|---|---|
| Random | 0.193 | 0.193 |
| PGM | $0.540 \pm 0.020$ | $0.450 \pm 0.020$ |
| DeepSet (NoSetFn) | $0.490 \pm 0.020$ | $0.316 \pm 0.008$ |
| DiffMF (ours) | $\mathbf{0.700 \pm 0.020}$ | $\mathbf{0.710 \pm 0.010}$ |
| EquiVSet$_{\text{ind}}$ (ours) | $0.590 \pm 0.010$ | $0.570 \pm 0.020$ |
| EquiVSet$_{\text{copula}}$ (ours) | $0.650 \pm 0.010$ | $0.600 \pm 0.010$ |

Table 10: Compound selection results with only the bioactivity filter.

| Method | PDBBind | BindingDB |
|---|---|---|
| Random | 0.099 | 0.009 |
| PGM | $0.910 \pm 0.010$ | $0.690 \pm 0.020$ |
| DeepSet (NoSetFn) | $0.910 \pm 0.010$ | $0.680 \pm 0.010$ |
| DiffMF (ours) | $0.920 \pm 0.010$ | $0.690 \pm 0.020$ |
| EquiVSet$_{\text{ind}}$ (ours) | $0.930 \pm 0.010$ | $0.697 \pm 0.006$ |
| EquiVSet$_{\text{copula}}$ (ours) | $\mathbf{0.931 \pm 0.008}$ | $\mathbf{0.700 \pm 0.008}$ |

preserving diversity, the task defined here only focuses on selecting the compounds with the highest bioactivity, which results a relatively simple selection process. The following is a detailed description.

**PDBBind:** To construct a data point $(V, S^*)$, we randomly sample 30 complexes as the ground set $V$ from the PDBBind database, and $S^*$ is generated by the five most active complexes in $V$. Finally, we obtain the training, validation, and test set with the size of $1,000, 100, 100$, respectively. **BindingDB:** We construct the ground set $V$ by randomly sampling 300 drug-targets from the BindingDB database and generate $S^*$ with the 15 most active drug-target pairs. We finally obtain the training, validation, and test set with the size of $10, 00, 1, 00$, and $1, 00$, respectively.

Table 10 shows that our methods outperform the baselines. Meanwhile, the baselines also show satisfactory results. That is because identifying the most active compounds is a relatively simple task, especially for the PDBBind dataset with complex structures. More specifically, the model could predict the activity value of complexes precisely without considering the interactions between elements in the set, since the structure of complexes has provided sufficient information for this task. It is worth noting that the models in this task perform better than that in Section 6 partly because a one-layer filter (i.e. bioactivity) represents an easier way to replicate the OS oracle than a two-layer filter (i.e. bioactivity and diversity). Nevertheless, both experimental results in Section 6 and here demonstrate the effectiveness of EquiVSet for facilitating the complicated compound selection process.

### F.5 Sensitivity Analysis of Hyperparameters

The proposed model EquiVSet$_{\text{copula}}$ has four important hyperparameters: the number of Monte Carlo sampling $m$ and mean-field iteration step $K$ in Algorithm 1, the rank of lower-rank perturbation $v$ in (22), and the temperature $\tau$ of Gumbel-Softmax in Algorithms 5 and 6. In this section, we discuss the impact of these hyperparameters through a sensitivity analysis on the Amazon product datasets.

**Impact of the Mean Field Iteration Step** Since iteration step $K$ controls the convergence of mean-field iterative algorithms, this hyperparameter is highly relevant to the final performance of EquiVSet$_{\text{copula}}$. We experiment with different $K$ on the Amazon product dataset. The results are shown in the first row of Figure 9. We notice that increasing K would degenerate the model's performance. This seems to be embarrassingly surprising at first glance, since a large stride $K$ encourages convergence with guarantee, resulting in a more robust training process. It is worth to be noted that in this method, we apply an amortized variational distribution to initialize the parameters for mean-field iterative algorithms. Since the amortized variational distribution is modeled with Gaussian copula, it can effectively capture the correlation among elements in the set, such that obtaining a better local optimal. However, if the iterative step $K$ is large, the model inclines to converge to the local optimal that is the same as the original mean-field iteration. As a result, the benefit of correlation-aware inference provided by the Gaussian copula would be diminished. This explains why the iterative step $K$ cannot be set too large.

**Impact of the Number of MC Sampling** The number of Monte Carlo (MC) sampling $m$ plays an important role in the proposed method. It is widely known that increasing number of samples would reduce the variance of MC sampling. Therefore, using larger $m$ would result in a better approximation of the gradient of multilinear extension $\nabla_\psi f_{\text{mt}}^{F_\theta}$ and thus better performance. This hypothesis is validated by the empirical results show in the second row of Figure 9. It can be seen that as the sample number increases, the performance rises steadily at first and then gradually converges

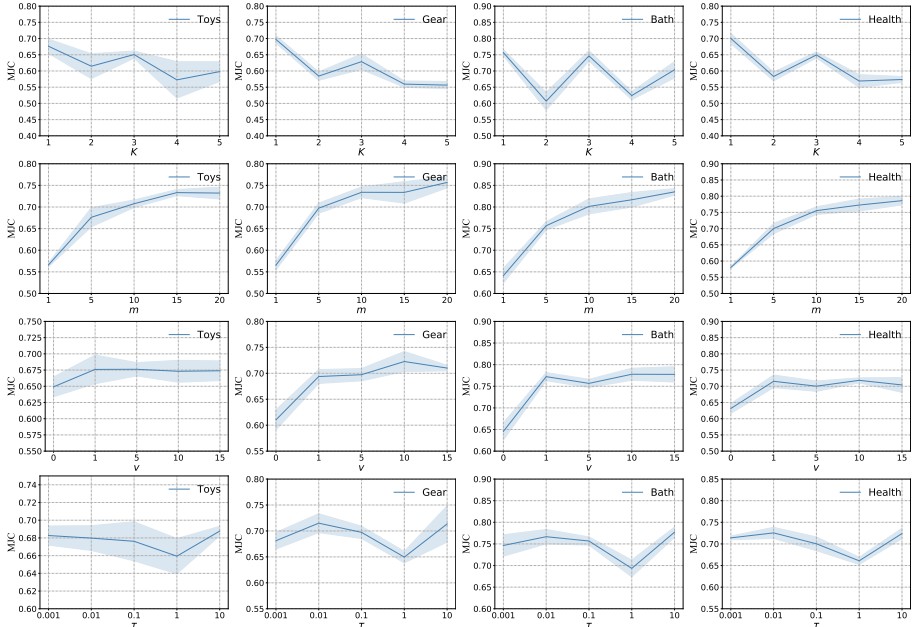

Figure 9: Sensitivity analysis of performance of $\texttt{EquiVSet}_{\text{copula}}$ under different hyperparameters (from top to bottom: the number of mean field iteration step $K$, the number of MC sampling $m$, the rank of lower-rank perturbation $v$, and the temperature of Gumbel-Softmax trick $\tau$).

into a certain level. Undoubtedly, a large number would increase the computational complexity. In this regard, we uniformly set it as $5$ in all experiments.

**Impact of the Lower-rank Perturbation**   Lower-rank perturbed covariance matrix enables the proposed method to model the correlation information of elements in the set. To investigate its impacts, we evaluate the performance of $\texttt{EquiVSet}_{\text{copula}}$ under different values of rank $v$. The results are demonstrated in the third row of Figure 9. Notably, the proposed model with $v = 0$ is equivalent to $\texttt{EquiVSet}_{\text{ind}}$. It can be seen that as the number of ranks increases, the performances also increase, indicating the hypothesis that employing the variational distribution with correlations can increase the model's representational capacity and thereby results in a better approximation in turn. It is worth noting that the most significant performance improvement is observed between the models with $v = 0$ and $v = 1$, and then as the value of $v$ continues to increase, the improvement becomes relatively small. This indicates that it is feasible to set the $v$ to a relatively small value to save computational resources while retaining competitive performance.

**Impact of the Temperature Parameter of Gumbel-Softmax**   The temperature parameter $\tau$ controls the trade-off between accuracy and variance of the approximation. With lower temperatures ($\tau \to 0$), the samples become more discrete but have a high variance of gradients. Alternatively, high temperatures ($\tau \to \infty$) result in smooth variables while enjoying a low variance of gradients. Fortunately, the experimental results in the last row of Figure 9 show that our model is quite robust with varying temperature values. It can be seen that the performance of models drops when $\tau = 1$, but the variance of performances is mild. We set $\tau = 0.1$ in the experiments.