# OpenReview forum: "Learning Neural Set Functions Under the Optimal Subset Oracle"
_NeurIPS.cc/2022/Conference — NeurIPS 2022 Accept_

### Official Review · Reviewer_HBfR · 2022-06-30

**Rating:** 8
**Confidence:** 4
**Soundness:** 4 excellent
**Presentation:** 4 excellent
**Contribution:** 4 excellent

**Summary:**

The authors tackle the task of set function learning under a weakly supervised setting, where the data points consist of sets and their optimal subsets. The proposed model combines an energy-based model with amortized variational inference to allow for several features (permutation invariance, varying grounds sets, a minimum prior approach, as well as scalability) and is further extended by a Gaussian copula variant to allow for correlation modelling. The model is evaluated on several data sets from different domains.

**Questions:**

-

**Limitations:**

Limitations and societal impact are discussed.

**Strengths And Weaknesses:**

## Strengths
- The overall model is well motivated. Each of the desired requirements for the model is discussed in detail and tackled in a principle manner.
- The overall storyline of the paper is well executed. The individual developmental steps and their problems are presented in a coherent structure.
- The paper includes a wide range of experiments.

## Weaknesses
- The reported decimal places in Table 1/2/3 suggest a measurement precision that is not valid given the reported standard deviations, which give us a precision of only the first two decimals for most reported settings.

## Other
- The number of runs and whether standard deviation or standard error are reported should be included in the caption of the tables. The runs are mentioned in the text, but the deviation vs error is not at all discussed (also not in the appendix).
- l302 claims "two real-world datasets". Why this specific formulation for MNIST/CelebA when the other experiments are also on real-world data?
- (very minor) l78 "delicate framework": The wording here feels suboptimal. It can be read in its negative connotation of being a rather fragile framework that requires great care to be trained. The rest of the paper does not suggest this to be the case, allowing for the positive meaning of the word. However, in the introduction, the reader does not know yet which of the two meanings in his/her mind the paper should be read.


### Typos
- l41 lacked $\to$ lacking

---

> ### Author Response · Authors · 2022-08-02
> **Answers to Reviewer**
>
> We thank the reviewer for the positive and extensive comments. We are refining the manuscript accordingly. Please stay tuned and see our answers below.
>
> ### Comments: Not valid given the reported standard deviations
>
> > The reported decimal places in Table 1/2/3 suggest a measurement precision that is not valid given the reported standard deviations, which give us a precision of only the first two decimals for most reported settings.
> >
>
> **ANSWER:** Thanks for pointing this out. It is true that the reported standard deviations make only the first two decimals valid for most reported results. However, in most cases, our approach with the worst performance (i.e., mean - stdev) still outperforms the baselines with the best performance (i.e., mean + stdev) significantly. This suggests the superiority of our approach. Hope this answer could address your concerns. If not, please feel free to let us know.
>
> ### Comments: The standard deviation vs error
>
> > The number of runs and whether standard deviation or standard error are reported should be included in the caption of the tables. The runs are mentioned in the text, but the deviation vs error is not at all discussed (also not in the appendix).
> >
>
> **ANSWER:** Sorry for the confusion. We repeat all experiments 5 times with different random seeds and report the mean and standard deviation of the results in Tables 1/2/3. We will add the corresponding statement in the revised paper.
>
> ### Comments: "Two real-world datasets”
>
> > l302 claims "two real-world datasets". Why this specific formulation for MNIST/CelebA when the other experiments are also on real-world data?
> >
>
> **ANSWER:** Sorry for the misleading. You’re right. We are misusing the term “real-world datasets” here. It shouldn’t be distinguished from the other experiments since the datasets used in the product recommendation and drug discovery are also “real-world datasets”. We will replace it with “two image datasets”.
>
> ### Comments: "delicate framework"
>
> > (very minor) l78 "delicate framework": The wording here feels suboptimal. It can be read in its negative connotation of being a rather fragile framework that requires great care to be trained. The rest of the paper does not suggest this to be the case, allowing for the positive meaning of the word. However, in the introduction, the reader does not know yet which of the two meanings in his/her mind the paper should be read.
> >
>
> **ANSWER:** Good suggestions! We ignore the negative connotation of “delicate”, which would exactly mislead readers to think that our framework is too fragile to be trained. Maybe the “elegant framework” is a better choice. We will update the manuscript accordingly.

---

> > ### Comment · Reviewer_HBfR · 2022-08-03
> > **Answer to the comments from the authors**
> >
> > Thank you for the detailed answers.
> >
> > > The reported decimal places in Table 1/2/3 suggest a measurement precision that is not valid given the reported standard deviations, which give us a precision of only the first two decimals for most reported settings.
> > ANSWER: Thanks for pointing this out. It is true that the reported standard deviations make only the first two decimals valid for most reported results. However, in most cases, our approach with the worst performance (i.e., mean - stdev) still outperforms the baselines with the best performance (i.e., mean + stdev) significantly. This suggests the superiority of our approach. Hope this answer could address your concerns. If not, please feel free to let us know.
> >
> > I should clarify my point from the review. My comment was not meant to doubt the significant difference between the results. There I completely agree with the authors. It was meant that given, as you agree, in most of the settings the 3/4th decimals do not convey any information, they should also not be reported. In its current form the tables, ignoring the stds, suggest that the experimental results have the precision to distinguish the mean performance between the models up to four digits. But the stds show that that is not the case.
> > Cutting them from the tables removes this misleading suggestion.

---

> > > ### Author Response · Authors · 2022-08-03
> > > **Answer to further comments**
> > >
> > > Huge thanks for your super quick reply and helpful suggestions. We feel sorry for misunderstanding your points before. Indeed, the mean of precision could distinguish the performance among different models. We will cut the standard deviations from the table in the revised paper.

---

> > > > ### Comment · Reviewer_HBfR · 2022-08-03
> > > > **Quick clarification**
> > > >
> > > > > We will cut the standard deviations from the table in the revised paper.
> > > > In case this was not just a typo, I only meant removing the insignificant digits from the table, not the complete standard deviations.

---

> > > > > ### Author Response · Authors · 2022-08-03
> > > > > **Answer to further comments**
> > > > >
> > > > > Thanks a lot for your patience and further suggestions.
> > > > >
> > > > > To better understand, Let's consider a specific case: in the current tables,  we report the result as $0.4414 ± 0.0036$. Do you mean we should rewrite it as $0.4410 ± 0.0036$ since the 4th decimal does not convey any information? Moreover, in the case of $0.2330 ± 0.0115$, it should be rewritten as $0.2300 ± 0.0115$, since the 3/4th decimals do not convey any information. Are we understanding correctly?

---

> > > > > > ### Comment · Reviewer_HBfR · 2022-08-03
> > > > > > **Digits**
> > > > > >
> > > > > > What I mean is to report in the case of your example $0.4414 \pm 0.0036$ as $0.441\pm 0.004$. The last decimal in $0.4414$ does not give any information here. As a thought experiment think about reporting a more extreme case, e.g., $0.12345678 \pm 0.2$. Here, everything after the first digit is noise and not stable information. For your second case we have $0.2330 \pm 0.0115$ as $0.23\pm 0.01$.
> > > > > >
> > > > > > (Of course if in a column e.g., all values are 0.123 and just one is 0.12 then it makes sense to report that last one as 0.120 to keep the visual structure of the table)

---

> > > > > > > ### Author Response · Authors · 2022-08-03
> > > > > > > **Answer to further comment**
> > > > > > >
> > > > > > > Thank you exactly! We will refine the manuscript accordingly.

---

### Official Review · Reviewer_6kuR · 2022-07-12

**Rating:** 8
**Confidence:** 4
**Soundness:** 3 good
**Presentation:** 3 good
**Contribution:** 3 good

**Summary:**

This paper proposes a learning framework for set functions. It combines a set of procedures and methods to obtain a unique set of properties for the learned function as listed in introduction.

**Questions:**

Please address questions about the experimental results, and the assumptions about the data distribution.

**Limitations:**

Yes, the discussion seems adequate.

**Strengths And Weaknesses:**

Strength:
This paper appears to be sound. It is well-written. Algorithms and procedures are clear.

Weaknesses:
My main concern are the limited experimental results, and also the assumptions made about the data distribution and their applicability for practice. Authors have not discussed whether their assumptions about the data applies to the datasets they have used in their experiments.

Experiments presented in Tables 2 and 3 seem to be limited. Can authors please explain how they chose these particular set of datasets? Results on more datasets could make the paper more convincing.

---

> ### Author Response · Authors · 2022-08-02
> **Answers to Reviewer (part 1)**
>
> Thanks for the helpful comments that help us to improve our work. Our specific answers are listed below.
>
> ### Comments: The assumptions made about the data distribution
>
> > My main concern are the limited experimental results, and also the assumptions made about the data distribution and their applicability for practice. Authors have not discussed whether their assumptions about the data applies to the datasets they have used in their experiments.
> >
>
> **ANSWER:** Thanks for the valuable comments.  You seem to be concerned about two things: i) the limited experiments, and ii) the assumptions about the data distribution. Here are our answers:
>
> 1. **Experimental results on the other two datasets, i.e., F-MNIST and CIFAR-10, also beat baselines.**
>
>     We have experimented on the other two datasets, i.e., F-MNIST and CIFAR-10. The results show that our approach still outperforms baselines significantly. Please find the details in the answer to the next question.
>
>     Notably, PGM [1] is the only previous work focusing on learning set functions under the optimal subset oracle. In PGM, only experiments for product recommendations (see table 2 in PGM) are performed. That’s to say, we have largely extended the application domains of set function learning under OS oracle. Not only on product recommendation (table 1), we also experiment on set anomaly detection (table 2), compound selection (table 3), and synthetic dataset (table 7).
>
> 2. **The assumption about underlying data distributions is very weak**
>
>     Sorry for the confusion. Generally speaking, for any scenario with the output being a subset $S$ of the given ground set $V$ of the input, the proposed model could be applied to predict the subset $S$ of the given ground set $V$. The only loose assumption is that the optimal subset oracle $S^*$ of a given ground set $V$ is generated by some underlying distribution formulated via a utility function that maximizes the utility value of OS oracle $S^*$ (see Eq.1 in the paper). We further assume the utility function could be parameterized by a deep neural network, thanks to the universal approximation theorem [2].
>
>     This assumption is very weak and generally makes sense in practice. We also apply this assumption to the datasets used in the experiments. Specifically, in the product recommendation, $V$ denotes the recommended products, and $S^*$ is the one the customer buys. Undoubtedly, the underlying generative distribution, or say the utility function is specified by the selection process of customers. In set anomaly detection, given a ground set $V$, $S^*$ is generated as the one containing anomaly data points. Therefore, the utility function in this setting is formulated as the anomaly pattern. Moreover, in the compound selection, we applied high bioactivity and diversity filters to select compounds. In this case, the utility function is determined by the bioactivity and diversity of the group of compounds.
>
>
> [1] Tschiatschek S, Sahin A, Krause A. Differentiable submodular maximization[J]. arXiv preprint arXiv:1803.01785, 2018.
>
> [2] Leshno M, Lin V Y, Pinkus A, et al. Multilayer feedforward networks with a nonpolynomial activation function can approximate any function[J]. Neural networks, 1993, 6(6): 861-867.

---

> > ### Author Response · Authors · 2022-08-02
> > **Answers to Reviewer (part 2)**
> >
> > ### Comments: The choice of the particular set of datasets
> >
> > > Experiments presented in Tables 2 and 3 seem to be limited. Can authors please explain how they chose these particular set of datasets? Results on more datasets could make the paper more convincing.
> > >
> >
> > **ANSWER:** Thanks for your helpful suggestions. Here are our answers:
> >
> > 1. **How we chose the particular set of datasets in Tables 2 and 3.**
> >
> >     We chose the double MNIST dataset since we think that it is an easy-to-understanding dataset to perform set anomaly detection. You can choose MNIST, but it only contains 10 different digits. It is arguably less challenging than the double MNIST, which contains 100 different digits. The CelebA is a widely-used dataset on set anomaly detection (see fig.3 in [3], fig.5 in [4], and fig.3 in [5]), partially because each image contains multiple labels, which makes it easy to construct the set anomaly  $S^*$  for a given ground set $V$.
> >
> >     As for compound selection, we chose this task since it is becoming increasingly more important in the field of AI aided drug discovery [6], and PDBBind and BindingDB are two well-known datasets used in this task. Previous works generally select compounds in several sequential steps, e.g., choosing the highly active compounds, then selecting diverse subsets from them, and finally excluding compounds that are bad for ADME [6]. The remaining compounds are the ones we select. In this paper, we can learn to conduct this complicated selection process in an end-to-end manner, given the OS supervision signals.  We think performing set prediction on compound selection is a brave attempt at AI for drug discovery that no one has considered before. Moreover, EquiVSet shows desirable gains on this task. We believe that it has potential influence on this new application.
> >
> > 2. **Results on more datasets make the paper more convincing**
> >
> >     Thanks for the helpful suggestions. We consider the other two datasets, i.e., F-MNIST and CIFAR-10. Similar to the CelebA dataset, we construct a set anomaly detection dataset using F-MNIST and CIFAR-10 respectively. Detailed settings will be placed in the revised appendix. We are updating the manuscript accordingly. The empirical results are shown in the following table:
> >
> >     |                        |       F-MNIST       |       CIFAR-10      |
> >     |:----------------------:|:-------------------:|:-------------------:|
> >     |         Random         |        0.1926       |        0.1926       |
> >     |           PGM          |   0.5433 ± 0.0193   |   0.4519 ± 0.0167   |
> >     |    DeepSet (NoSetFn)   |   0.4854 ± 0.0185   |   0.3161 ± 0.0080   |
> >     |      DiffMF (Ours)     | **0.7021 ± 0.0252** | **0.7114 ± 0.0116** |
> >     |   EquiVSet_ind (Ours)  |   0.5931 ± 0.0098   |   0.5685 ± 0.0224   |
> >     | EquiVSet_copula (Ours) |   0.6457 ± 0.0108   |   0.6003 ± 0.0122   |
> >
> >     We see that the variants of our model consistently outperform baseline methods strongly. Moreover, DiffMF seems to perform better than EquiVSet_ind and EquiVSet_copula in set anomaly detection (similar results can be found in table 2 in the paper). However, this is not a consistent phenomenon. It seems that in most scenarios, e.g., product recommendation, compound selection, and synthetic dataset, EquiVSet performs better than DiffMF.
> >
> > [3] Zaheer M, Kottur S, Ravanbakhsh S, et al. Deep sets[J]. Advances in neural information processing systems, 2017, 30.
> >
> > [4]Lee J, Lee Y, Kim J, et al. Set transformer: A framework for attention-based permutation-invariant neural networks[C]//International conference on machine learning. PMLR, 2019: 3744-3753.
> >
> > [5] Zhang D W, Burghouts G J, Snoek C G M. Set prediction without imposing structure as conditional density estimation[J]. arXiv preprint arXiv:2010.04109, 2020.
> >
> > [6] Gimeno A, Ojeda-Montes M J, Tomás-Hernández S, et al. The light and dark sides of virtual screening: what is there to know?[J]. International journal of molecular sciences, 2019, 20(6): 1375.

---

> > > ### Comment · Reviewer_6kuR · 2022-08-04
> > > **Assumptions**
> > >
> > > Thank you for your thoughtful response. It is convincing to see the new results. Overall, the improvements in the paper address my concerns and I will increase my score for the paper.
> > >
> > > Please consider incorporating your clarifications in the paper, e.g., the points about your assumptions on the data distribution.

---

> > > > ### Author Response · Authors · 2022-08-05
> > > > **Thanks for your favor in accepting our work**
> > > >
> > > > Thanks a lot for reading our responses and accepting our work. We are refining the paper accordingly. The assumption of the data distribution and the new empirical results will be included.

---

### Official Review · Reviewer_vcKY · 2022-07-20

**Rating:** 7
**Confidence:** 4
**Soundness:** 3 good
**Presentation:** 3 good
**Contribution:** 3 good

**Summary:**

This paper introduces EquivSet, an algorithm for learning set functions that satisfies the following desiderata: permutation invariance, varying ground set, minimum prior and scalability. EquivSet learns set functions under the optimal subset oracle using a maximum likelihood paradigm.  Specifically, the authors use an energy based model(EBM) to define the set mass function. The usage of EBM satisfies the minimum prior requirement.  A DeepSets style architecture is also used to satisfy the permutation invariance constraint on set functions and to also handle sets of arbitrary cardinality. However, learning EBMS over sets introduces some difficulties which the authors alleviate by proposing a mean field variational inference approach in the style of variational auto encoders. Amortization is also used to ensure scalability.

**Questions:**

My main question has to do with how Set Transformer performs compared to EquivSet as detailed above.

**Limitations:**

Limitations are sufficiently outlined in the Limitations and Broader Impact section.

**Strengths And Weaknesses:**

The proposed method is well motivated and tackles the set representation learning problem from a perspective not yet considered.  The EMB approach and the variational inference approach introduce more complexity compared to the well used DeepSets and Set Transformer models.

Additionally, Set Transformer, which is trained similarly to DeepSets, is not used as a baseline. Set Transformer is a simple baseline that normally outperforms DeepSets and is easy to train. Hence i recommend the authors to include it in the baselines.

Finally, it seems the SAB and ISAB layers of Set Transformer can be used in the model in place of the DeepSets style architecture. Many set representation papers show that these attention based backbones perform much better than DeepSets.

**The authors have answered my questions satisfactorily hence i increase my initial score.**

---

> ### Author Response · Authors · 2022-08-02
> **Answers to Reviewer (part 1)**
>
> Thanks a lot for your valuable comments. Your suggestions are very helpful for further improving the work, and we are refining the manuscript accordingly.
>
> ### Comments: EquiVSet is more complex than Set Transformer
>
> > The EMB approach and the variational inference approach introduce more complexity compared to the well used DeepSets and Set Transformer models.
> >
>
> **ANSWER:** Thanks for the comment. This is a misunderstanding regarding the OS oracle studied in this paper. It is noteworthy that both the DeepSet and Set Transformers can serve as backbones of our approach. However, they cannot work as a standalone solution in the OS oracle since no set function value is available. Their adapted version (i.e., DeepSet(NoSetFn) and Set Transformer(NoSetFn), adapted to serve as the amortized networks in EquiVSet) could work as a baseline, which can be viewed as merely modeling the amortized network in our EquiVSet framework. Therefore, DeepSet and Set Transformer do not learn a set function explicitly under the OS oracle, although they can be adapted to our empirical studies.
>
> Moreover, to model the EBM  $p_\theta(S|V) \propto \exp(F_\theta (S;V))$, one could take DeepSet or Set Transformer as backbones, i.e., the utility function $F_\theta$. More advanced architectures might bring additional performance gains, but it is not the main focus of this paper. Therefore, we use the simplest one, i.e., DeepSet in both EquiVSet and all baselines, for fair comparisons.
>
> Furthermore, we have also experimented with the Set Transformers backbone. The results are presented in the answer to the next comment.
>
> ### Comments: Missing the baseline of Set Transformer
>
> > Additionally, Set Transformer, which is trained similarly to DeepSets, is not used as a baseline. Set Transformer is a simple baseline that normally outperforms DeepSets and is easy to train. Hence i recommend the authors to include it in the baselines.
> >
>
> **ANSWER:** Thanks for the suggestions. Here we consider including SetTransfomers in the baselines. Note that, same as DeepSet (NoSetFn), the SetTransformers baseline also does not learn a set function explicitly, though it could be adapted and applied to our empirical studies. For fair comparisons, we also replace the backbone of EquiVSet with SetTransformers. Here are the empirical results:
>
> |           | SetTransformer (NoSetFn) |        DiffMF       |     EquiVset_ind    |   EquiVset_copula   |
> |:---------:|:------------------------:|:-------------------:|:-------------------:|:-------------------:|
> |    Toys   |      0.6397 ± 0.0256     |   0.6940 ± 0.0278   |   0.6803 ± 0.0179   | **0.7167 ± 0.0060** |
> | Furniture |    **0.1748 ± 0.0078**   |   0.1689 ± 0.0164   |   0.1587 ± 0.0062   |   0.1656 ± 0.0071   |
> |    Gear   |      0.6390 ± 0.0055     | **0.7532 ± 0.0214** |   0.6876 ± 0.0241   |   0.6970 ± 0.0105   |
> |  Carseats |      0.2187 ± 0.0054     |   0.2191 ± 0.0059   | **0.2193 ± 0.0086** |   0.2159 ± 0.0083   |
> |    Bath   |      0.7248 ± 0.0054     |   0.7989 ± 0.0180   |   0.7997 ± 0.0111   | **0.8104 ± 0.0133** |
> |   Health  |      0.6844 ± 0.0113     |   0.7468 ± 0.0207   |   0.7428 ± 0.0185   | **0.7599 ± 0.0161** |
> |   Diaper  |      0.7885 ± 0.0046     |   0.8712 ± 0.0093   |   0.8714 ± 0.0121   | **0.8859 ± 0.0086** |
> |  Bedding  |      0.7646 ± 0.0196     |   0.8587 ± 0.0081   |   0.8579 ± 0.0214   | **0.8607 ± 0.0066** |
> |   Safety  |      0.2565 ± 0.0049     |   0.2400 ± 0.0057   |   0.2417 ± 0.0146   | **0.2649 ± 0.0254** |
> |  Feeding  |      0.7833 ± 0.0057     | **0.8856 ± 0.0037** |   0.8810 ± 0.0097   |   0.8775 ± 0.0088   |
> |  Apparel  |      0.6819 ± 0.2325     |   0.7622 ± 0.0123   |   0.7535 ± 0.0144   | **0.7666 ± 0.0119** |
> |   Media   |      0.5389 ± 0.0189     |   0.6151 ± 0.0075   |   0.6094 ± 0.0117   | **0.6196 ± 0.0094** |
>
> It shows that our models with the Set Transformer backbone outperform the Set Transformer(NoSetFn) consistently. One could also compare the results of table 1 in the paper. It can be seen that the proposed EquiVSet (with the DeepSet backbone) also performs better than the Set Transformer baseline. Detailed discussions will be placed in the revised appendix.

---

> > ### Author Response · Authors · 2022-08-02
> > **Answers to Reviewer (part 2)**
> >
> > ### Comments: The usage of SAB and ISAB layers of the Set Transformer
> >
> > > Finally, it seems the SAB and ISAB layers of Set Transformer can be used in the model in place of the DeepSets style architecture. Many set representation papers show that these attention based backbones perform much better than DeepSets.
> > >
> >
> > **ANSWER:** Great suggestions! We have conducted corresponding experiments. Here we consider two SAB blocks with two multi-head attentions. Detailed settings will be placed in the revised appendix. The results are shown in the table of the previous answer. It shows that EquiVSet with SetTransformer backbone could bring significant performance gains, compared with the DeepSet backbone. It also tells us that better architecture of set functions would result in better performance. We think it is a potential direction to improve our work. Thank you exactly!

---

### Official Review · Reviewer_hn9G · 2022-07-20

**Rating:** 6
**Confidence:** 4
**Soundness:** 2 fair
**Presentation:** 2 fair
**Contribution:** 2 fair

**Summary:**

The paper addresses the problem of learning set functions. There are two variations of this problem, one is with function value (FV) oracle in which the supervised data takes the form of (set S_i, function value f_i of set S_i) and the goal is to learn the function mapping. One has to gather this info for large number of sets, making this process prohibitively expensive and often the FV oracle is not feasible. The main focus of the paper is on the variation of the problem with Optimal Subset (OS) oracle, wherein the data takes the form of $(V_i, S_i^*)$ pairs where $V_i $ is subset of $V$ (global candidate set) and $S_i^*$ is the optimal subset of $V_i$ that maximizes the utility. This setup implicitly captures the FV oracle and is more practical.

Given such data from the OS oracle, they propose a method based on variational inference to learn the mapping.  The first step is to cast the problem as maximum likelihood estimation by replacing the utility function $F_{\theta}(S,V)$ with a probability distribution $P_{\theta}(S|V)$ such that the probability of S given V is proportional to the utility. They list some basic (natural ) properties that such a distribution should satisfy a) permutation invariance, b) varying ground set etc. Further they want to have minimum prior assumption and scalable learning algorithm.

They give a distribution based on energy-based modeling (EBM) which satisfies the aforementioned properties and admits efficient training and inference. Directly learning with the EBM is difficult for reasons such as intractable partition function, to circumvent the difficulty they propose an approximate maximum likelihood learning using variational approximation of $P_{\theta}(S|V)$ by product of independent Bernoulli distributions.

The method is evaluated on production recommendation, set anomaly detection, double MNIST, CelebA, Compund selection in AI-aided Drug Discovery and other real world datasets.
The results show that the proposed method significantly outperforms the baselines for learning set functions.


**Questions:**

Please address the weaknesses.


**Limitations:**

yes

**Strengths And Weaknesses:**

Strengths:
S1: It studies the problem of learning set functions under the OS oracle, which is practically important but has been addressed by limited number of works.

S2: The reduction of this problem to maximum likelihood and later to variational approximation is novel and makes it amenable to application of this well-developed machinery.

S3: The empirical evaluation is extensive and beats the closely related baseline comprehensively.

Weaknesses:
W1: There two steps of approximation that introduce their own approximation error. It is not clear how these are propagating and affecting the eventual solution. I believe with more approximation layers the error (sub-optimality) is higher.

W2: Lack of theoretical results to shed light on the sub-optimality of the proposed algorithm. I think, the paper can be much better with these.

---

> ### Author Response · Authors · 2022-08-02
> **Answers to Reviewer (part 1)**
>
> Thanks for your insightful comments and helpful suggestions! Our answers are listed below.
>
> ### Comments: The approximation error
>
> > There two steps of approximation that introduce their own approximation error. It is not clear how these are propagating and affecting the eventual solution. I believe with more approximation layers the error (sub-optimality) is higher.
> >
>
> **ANSWER:** Thanks a lot for your valuable comments that help us to improve our work. It seems that you are curious about the error accumulation of the two-step approximation, which if we understand correctly is: i) the maximum likelihood formulation, in which we cast the problem into MLE by replacing $F_\theta (S;V)$ with a probability $p_\theta(S|V)$; and ii) the variational approximation of EBMs. To better answer this question, we cast it into two separate parts:
>
> 1. **The MLE formulation is an affordable paradigm to relax the original problem**
>
>     Given the data generative distribution $\mathbb{P}(V,S^*) $, the set function learning problem under OS oracle is to find an optimal model $\theta^* \in \mathcal{H}, s.t. S^* = \arg\max_{S\in 2^V} F_{\theta^*} (S;V),  \forall (V, S^*) \sim \mathbb{P}(V,S^*) $.  A natural design choice is applying some combinatorial algorithm, say, the greedy algorithm for maximizing the utility,  which would arrive at the inferred subset $\hat{S}$ , then calculate some divergence measure between $\hat{S}$ and $S^*$. However, in general it is NP-hard to solve the combinatorial problem. Furthermore, the greedy algorithm renders the process non-differentiable.
>
>     To resolve the above issue, we cast this problem into the maximum likelihood estimation: $\hat{\theta} = \arg\max_{\theta} \mathbb{E}_{\mathbb{P}(V,S)} [\log p_\theta (S|V)]$ such that the probability $p_\theta$  is proportional to the utility value. As you said, the MLE  would introduce some approximation errors, and bounding this error (i.e., $p(|\hat{\theta}-\theta^*|)$) is a long-lasting open problem. We still work on it and leave it as an important future work.
>
> 2. **The variational inference is a practical surrogate to approximate the likelihood of EBMs**
>
>     It is true that inference on EBMs would unavoidably introduce approximation errors, since exactly evaluating the likelihood of EBMs is #P-complete [1]. There are a lot of works focusing on how to perform inference on EBMs, such as the Langevin MCMC applied in the set generation tasks [2]. However, the MCMC method is arguably more computationally complex than the variational methods. Thereby, we advocate variational inference here, which is more practical and efficient in the set function learning scenarios, but it is also worth exploring other methods to train EBMs over sets.
>
>
> Overall, approximation errors do accumulate in both approximation layers. However, from the practical perspective, as you said: “*the reduction of this problem to maximum likelihood and later to variational approximation is novel and makes it amenable to application of this well-developed machinery*.” Considering how to reduce approximation errors for each step would significantly improve our work. We leave it as important future work.
>
> [1] Valiant L G. The complexity of computing the permanent[J]. Theoretical computer science, 1979, 8(2): 189-201.
>
> [2] Zhang D W, Burghouts G J, Snoek C G M. Set prediction without imposing structure as conditional density estimation[J]. arXiv preprint arXiv:2010.04109, 2020.

---

> > ### Author Response · Authors · 2022-08-02
> > **Answers to Reviewer (part 2)**
> >
> > ### Comments: Lack of theoretical results
> >
> > > Lack of theoretical results to shed light on the sub-optimality of the proposed algorithm. I think, the paper can be much better with these.
> > >
> >
> > **ANSWER:** Thank you for the helpful suggestions. Unfortunately, it is a long-lasting open problem to prove the sub-optimality of the proposed algorithm due to the two-step approximation. It also leaves a large space for future exploration and we are still working on it.
> >
> > Moreover, we want to further claim that the proof of the sub-optimality of the second step approximation is also a long-lasting open problem. The proof of the sub-optimality of the marginal-based loss (i.e., Eq.5 in the paper) is nontrivial. Even in its original paper (i.e., Domke’s work [3]), the sub-optimality of this objective isn’t well discussed.
> >
> > However, such an in-accurate learning framework could bring some benefits. As pointed out by Justin Domke in his work [3], this objective benefits from taking the approximation errors of inference algorithm into account while learning. Hanjun Dai et al [4] also apply this objective to learn Markov Random Fields for graph classification. Similar ideas are also applied to other works [5,6]. The empirical results in our work further confirm its practicality.
> >
> > [3] Domke J. Learning graphical model parameters with approximate marginal inference[J]. IEEE transactions on pattern analysis and machine intelligence, 2013, 35(10): 2454-2467.
> >
> > [4] Dai H, Dai B, Song L. Discriminative embeddings of latent variable models for structured data[C]//International conference on machine learning. PMLR, 2016: 2702-2711.
> >
> > [5] Zheng S, Jayasumana S, Romera-Paredes B, et al. Conditional random fields as recurrent neural networks[C]//Proceedings of the IEEE international conference on computer vision. 2015: 1529-1537.
> >
> > [6] Belanger D, Yang B, McCallum A. End-to-end learning for structured prediction energy networks[C]//International Conference on Machine Learning. PMLR, 2017: 429-439.

---

### Official Review · Reviewer_3MWU · 2022-07-22

**Rating:** 7
**Confidence:** 2
**Soundness:** 2 fair
**Presentation:** 3 good
**Contribution:** 3 good

**Summary:**

The paper proposes a way to learn set functions (ie. find a subset $S \subseteq V$ that maximizes some utility function $F_\theta$ that we want to learn) when we are only giving examples of optimal subsets $S^*_i \subseteq V_i$ from an optimal subset oracle.

This is different to other works that learn from function value oracles that provide actual utility values $f_i$ for specific subsets $S_i$.

The paper proposes a permutation-invariant architecture. It makes the problem tractable through mean-field variational inference and further amortizes inference via an additional neural network.

It reports great empirical results compared to several baselines.

---
Thanks for addressing my questions and updating the paper!

**Questions:**

1. I believe the max entropy proof for the EBM (App B.1.) can be simplified considerably. In the case of $S \mid V$, $S$ can be simply treated as a categorical with $2^{|V|}$ categories, and the softmax categorical distribution (EBM) is the exponential family distribution for that and maximum entropy as is well-known. This works because at this point, the internal structure of $S$ is not important. It's only when we look at different $V$ that you want to take its structure into account. Is this true?

**Limitations:**

The "broader impact" should be removed from the "Limitations & Broader Impact" paragraph title because it does not address broader societal impact (which is okay, but the title is misleading).

**Strengths And Weaknesses:**

This was requested as an emergency review. I will try my best to offer a meaningful review (ofc), but I'm very much looking forward to discussing with the other reviewers.

The approach seems original (I am not very well acquainted with prior literature in this area though). The quality and clarity of the paper is great overall, and it seems like a significant contribution.

I very much like the argument that it is easier to acquire data for the optimal subset oracle than the usual function value oracle in many practical cases as trying to obtain calibrated utility values from human labellers sounds improbable. Thus, I think this idea is of great practical importance.

The empirical validation looks sensible.

I recommend acceptance with low confidence given the nature of this review and that my background is not aligned with the subfield.

---

> ### Author Response · Authors · 2022-08-02
> **Answers to Reviewer**
>
> Thanks for your positive affirmation for our work. Here are our answers to your questions.
>
> ### Comments: Simplify the max entropy proof
>
> > I believe the max entropy proof for the EBM (App B.1.) can be simplified considerably. In the case of S∣V, S can be simply treated as a categorical with 2^|V| categories, and the softmax categorical distribution (EBM) is the exponential family distribution for that and maximum entropy as is well-known. This works because at this point, the internal structure of S is not important. It's only when we look at different V that you want to take its structure into account. Is this true?
> >
>
> **ANSWER:** Thanks for your insightful comments! It is true that $p_\theta(S|V)$ maximizes the entropy if we simply regard it as a categorical with $2^{|V|}$ categories, since the exponential family distribution is the well-known maximum-entropy distribution. However, it is noteworthy that the exponential family (e.g., the categorical distribution) is a specific type of EBMs. As you said, we cannot simply restrict EBM as a categorical which has fixed categories, since we look at different $V$, i.e., varying ground sets. Therefore, we consider a more general EBM with the form of $\exp(F_\theta (S;V))/Z$. To prove it is the maximum entropy distribution, we reproduce and adapt the proof from Jaynes [1] in the context of set function learning.
>
> Hope this answer could address your concern. If not, please feel free to let us know. We would be glad to answer any further questions you may have.
>
> [1] Jaynes E T. Information theory and statistical mechanics[J]. Physical review, 1957, 106(4): 620.
>
> ### Comments: Misuse of the term "broader impact”
>
> > The "broader impact" should be removed from the "Limitations & Broader Impact" paragraph title because it does not address broader societal impact (which is okay, but the title is misleading).
> >
>
> **ANSWER:** Thank you for pointing this out. We will remove the “broader impact” from the title and update the manuscript accordingly.

---

### Author Response · Authors · 2022-08-02
**Summary (part 1)**

We thank all reviewers for their favor of accepting our work, and also thank for their constructive and extensive comments that help us to improve this work.

We would first like to summarize the paper according to the reviewers:

- Our work proposes a way to learn set functions (i.e., find a subset $S\subseteq V$ that maximizes the utility function $F_\theta$ that we want to learn) when the optimal subsets $S^*_i \subseteq V_i$ are given from an optimal subset oracle. This setting is different to other works that learn set functions from the function value oracle that provide utility values $F_i$ for each specific subset $S_i$. Thus this setting is arguably more practically important but is surprisingly overlooked by previous works.

    > Reviews agree that this contribution is significant: R****3MWU**** “*I very much like the argument that it is easier to acquire data for the optimal subset oracle than the usual function value oracle in many practical cases as trying to obtain calibrated utility values from human labellers sounds improbable. Thus, I think this idea is of great practical importance*.”; R****hn9G**** “*It studies the problem of learning set functions under the OS oracle, which is practically important but has been addressed by limited number of works.*”; R****vcKY**** “*the proposed method is well motivated and tackles the set representation learning problem from a perspective not yet considered.*”
    >
- To learn set functions under the optimal subset oracle, we propose to cast the problem into maximum likelihood estimation by replacing the utility function $F_\theta(S;V)$ with an energy-based model $p_\theta(S|V)$ such that it is proportional to the utility value, satisfies some desiderata for set functions (e.g., permutation invariance, etc). Then mean-field variational inference and its amortized variants are proposed to learn EBMs on the sets.

    > Reviews agree that this approach is novel: R****3MWU**** “*the approach seems original. The quality and clarity of the paper is great overall, and it seems like a significant contribution.*”; R****hn9G**** “*The reduction of this problem to maximum likelihood and later to variational approximation is novel and makes it amenable to application of this well-developed machinery.*”; R****HBfR**** “*Each of the desired requirements for the model is discussed in detail and tackled in a principle manner*”.
    >
- We evaluate our approach in a wide range of applications, including product recommendation, set anomaly detection, and compound selection in AI-aided drug discovery. The empirical results show our approach is promising.

    > Reviews agree on the soundness of our work: R****3MWU**** “*the empirical validation looks sensible*”; R****hn9G**** “*the empirical evaluation is extensive and beats the closely related baseline comprehensively*”; R****HBfR**** “*the paper includes a wide range of experiments*”.
    >

---

> ### Author Response · Authors · 2022-08-02
> **Summary (part 2)**
>
>
> Meanwhile, we summarize two main **concerns** from the reviewers here and leave detailed answers to the replies to each reviewer. We would be glad to answer any further questions you may have after reviewing the answers.
>
> 1. **Comparison to other baselines.**
>
>     > R****vcKY**** suggests including the Set Transformers as baselines.
>     >
>     - We have included the Set Transformer as new backbone and its adapted version (Set Transformer(NoSetFn) ) as a baseline. The results (please check [the details](https://openreview.net/forum?id=GXOC0zL0ZI&noteId=CjkyPQ9enzd) in the answer to R****vcKY****) show that our approach outperforms it comprehensively.
>     - We want to clarify that both the DeepSet and Set Transformers serve as  backbones of our approach.  They cannot work as a standalone solution in the OS oracle since no set function value is available. Their adapted version (i.e., DeepSet(NoSetFn) and  Set Transformer(NoSetFn),  adapted to serve as the amortized networks in EquiVSet) could work as a baseline, which can be viewed as merely modelling the amortized network in our EquiVSet framework.  More advanced architectures might bring additional performance gains, but it is not the major focus of this paper. Therefore, we use the simplest one, i.e., DeepSet, in both our approach and all baselines for fair comparisons. Moreover, further experiments with the Set Transformers backbone show that our approach has great potential to be improved with more advanced architecture.
> 2. **Experiment with more datasets.**
>
>     > R****6kuR**** suggests experimenting on more datasets.
>     >
>     - We have experimented on the other two datasets, i.e., F-MNIST and CIFAR-10, on set anomaly detection. The results (please check [the details](https://openreview.net/forum?id=GXOC0zL0ZI&noteId=QCRqoJkBQu) in the answer to R****6kuR****) show that our approach still outperforms baselines significantly.
>     - It is noteworthy that, in PGM, the only existing work for set function learning under the OS oracle, merely the experiment of product recommendation is conducted. We have widely extended the application domain of set function learning under the optimal subset oracle. In addition to product recommendations, we also experiment with set anomaly detection, compound selection, and synthetic datasets. The empirical results consistently confirm the superiority of our approach.

---

### Author Response · Authors · 2022-08-05
**Main changes marked in blue color**

Dear Reviewers,

Thanks a lot for your favor in accepting our work, and thanks for your helpful and constructive comments. We have revised the manuscript accordingly. The main changes are marked in blue. Here are they:

- Appendix E.4 for R**6kuR:** "*assumption on the data distribution*";
- Appendix F.3 for R***vcKY:*** “*comparisons with Set Transformer*”;
- Appendix F.4 for R**6kuR:** "results *on more datasets, i.e., F-MNIST & CIFAR-10*";
- Proper results demonstration in all tables for R***HBfR**: “The measurement precision is not valid given the reported standard deviations”.

Best,

The authors

---

### Meta-Review · Area_Chair_3CeT · 2022-08-26

**Recommendation:** Accept
**Confidence:** Certain

**Metareview:**

Reviewers have expressed strongly in favour of acceptance, two improving their score after the rebuttal and discussion. I’m happy to recommend acceptance.

**Award:**

No

---

### Decision · Program_Chairs · 2022-09-14

Accept